# Patient-derived pancreas-on-a-chip to model cystic fibrosis-related disorders

Kyu Shik Mun[1], Kavisha Arora[1], Yunjie Huang[1], Fanmuyi Yang[1], Sunitha Yarlagadda[1], Yashaswini Ramananda[1,2], Maisam Abu-El-Haija[3,4], Joseph J. Palermo[3,4], Balamurugan N. Appakalai[5], Jaimie D. Nathan[6] & Anjaparavanda P. Naren[1]

Cystic fibrosis (CF) is a genetic disorder caused by defective CF Transmembrane Conductance Regulator (CFTR) function. Insulin producing pancreatic islets are located in close proximity to the pancreatic duct and there is a possibility of impaired cell-cell signaling between pancreatic ductal epithelial cells (PDECs) and islet cells as causative in CF. To study this possibility, we present an in vitro co-culturing system, pancreas-on-a-chip. Furthermore, we present an efficient method to micro dissect patient-derived human pancreatic ducts from pancreatic remnant cell pellets, followed by the isolation of PDECs. Here we show that defective CFTR function in PDECs directly reduced insulin secretion in islet cells significantly. This uniquely developed pancreatic function monitoring tool will help to study CF-related disorders in vitro, as a system to monitor cell-cell functional interaction of PDECs and pancreatic islets, characterize appropriate therapeutic measures and further our understanding of pancreatic function.

[1] Department of Pediatrics, Division of Pulmonary Medicine, Cincinnati Children's Hospital Medical Center, Cincinnati, OH 45229, USA. [2] Department of Medical Biotechnology, University of Illinois, Rockford, IL 61107, USA. [3] Division of Pediatric Gastroenterology, Hepatology and Nutrition, Cincinnati Children's Hospital Medical Center, Cincinnati, OH 45229, USA. [4] Department of Pediatrics, University of Cincinnati, Cincinnati, OH 45229, USA. [5] Department of Surgery, Center for Cellular Transplantation, Cardiovascular Innovation Institute, University of Louisville, Louisville, KY 40222, USA. [6] Division of Pediatric General and Thoracic Surgery, Cincinnati Children's Hospital Medical Center, Cincinnati, OH 45229, USA. Correspondence and requests for materials should be addressed to J.D.N. (email: jaimie.nathan@cchmc.org) or to A.P.N. (email: anaren@cchmc.org)

The cystic fibrosis transmembrane conductance regulator (CFTR) protein is located on the apical membrane of epithelial cells in multiple organs, including lung, sweat gland, gastrointestinal tract, and pancreas, and its dysfunction is responsible for the clinical manifestations of cystic fibrosis (CF)[1–4]. CFTR is a cyclic AMP (cAMP)-dependent chloride and bicarbonate transport channel and plays an important role in maintaining salt and water balance on the epithelial surface. Defective CFTR channel function lowers the water content in the lumen, which leads to the development of a thick and viscous mucus on the epithelial surfaces in CF-affected organs[3]. To date, more than 2000 CFTR mutations have been identified since the first discovery of CF in 1938[5]. CFTR mutations are classified into six categories according to the primary molecular defect of the CFTR protein: synthesis (class I), trafficking process (II), gating (III), conductance (IV), mRNA stability (V), and CFTR stability (VI)[6].

CF-related diabetes (CFRD) is a frequent and deadly complication in CF. A patient with CF has an increasing risk of developing diabetes with age of 5% per year, reaching 50% by age 40s[7,8]. CFRD affects 2% of children, 19% of adolescents, and as high as 50% of adults[7]. Glucose imbalance due to CFRD has been correlated with increased morbidity and mortality in patients with CF. This calls for a need to develop approaches to study CFRD and identify therapeutic measures to potentially manage disordered glucose metabolism in CFRD. CFRD is complex as it exhibits the features of both the lack of insulin typical of type 1 diabetes (T1D) and the insulin resistance typical of type 2 diabetes (T2D)[9]. Whether a lack of CFTR function in CF patients directly manifests into CFRD remains unclear. Patients with CFRD show more severe side effects with significant loss of lung function and imbalanced nutrition than CF patients without diabetes[10]. CFTR is highly expressed in the pancreatic ductal epithelial cells (PDECs)[1,11–13], which are located in close proximity to pancreatic islets[14]; however,. the functional relationship between these two cell types in CFRD remains unclear. To investigate the role of CFTR in CFRD and functional correlation between PDECs and islet cells, we isolated PDECs and pancreatic islets from pancreatitis patients, who underwent total pancreatectomy with islet autotransplantation (TPIAT)[15]. We cultured these cell types in a microfluidic device to develop pancreas-on-a-chip, an in vitro model system to mimic the functional interface between PDECs and pancreatic islets. The pancreas-on-a-chip allows us to monitor cell–cell functional interaction directly and efficiently using only a small number of patient-derived cells. The microfluidic device was developed in 1979 as a miniature gas chromatograph[16] and it has been exponentially innovated in functionality and design. In recent decades, microfluidics have been used as an in vitro model system for cell culture[17–19], because of its high reproducibility, ability to mimic function and structure of organs, and some unique applications such as real-time PCR[20], single-cell western blot[21], wearable sensor[22], and organ-on-a-chip[23,24]. Pancreas-on-a-chip will facilitate elucidating the mechanism of cross-talk between PDEC and islet cells, which is key to understanding the relationship between CF and diabetes. Classified mutations in the CFTR gene may show variable pathological processes in the development of CFRD.

Here, we have successfully co-cultured patient-derived PDECs and islet cells in the same chip and observed that attenuating CFTR function in PDECs reduces insulin secretion in islet cells by 54%. This pancreas-on-a-chip is an innovative approach of developing personalized medicine to address heterogeneity in CFRDs.

## Results

**Isolation of patient-derived PDECs and pancreatic islets.** Pancreatitis patients who have a debilitating course of acute recurrent or chronic pancreatitis may undergo TPIAT) to relieve their pain and incapacitation, as shown in Fig. 1a. During TPIAT, the pancreas is resected and digested to isolate pancreatic islets. The islets are infused into the liver through the portal vein to engraft within the hepatic sinusoids and maintain their endocrine function. The pancreatic remnant cell pellet was obtained after islet cell isolation (Fig. 1b) and shown to contain pancreatic islets, acinar cells, and PDECs) (Fig. 1c). Pancreatic islets in the pancreatic remnant cell pellet were visualized by adding dithizone solution, which turns the color of islets to red (Fig. 1d)[25]. The clustered red-colored pancreatic islets were manually isolated and cultured in vitro (Fig. 1g). The diameter of the isolated pancreatic islets ranged from 50 to 300 μm.

From the pancreatic remnant cell pellet, we successfully isolated the pancreatic duct with diameter ranging from 90 to 1000 μm by microdissection under a stereo microscope (Fig. 1e). Hematoxylin and eosin (H&E) staining of the pancreatic duct demonstrated that the predominant cells, PDECs, were surrounded by collagen and connective tissue (Fig. 1f). Pancreatic ducts were enzymatically digested to separate the cells (Supplementary Fig. 1). Isolated pancreatic ductal cells were subsequently embedded in Matrigel and clustered into small spherical structures after isolation (day 0) and formed organoid structures at day 1, with a luminal fluid-filled area in the center (Fig. 1h). The organoids grew into larger spheres with a diameter of approximately 400 μm at day 6 from isolation. Growth of organoids could exceed 3 mm in diameter. Organoids in Matrigel-formed duct-like structures when they contacted the surface of the substrate and migrated out forming a monolayer (Fig. 1i). Although the mechanism is unclear, we have consistently observed this phenomenon (Supplementary Fig. 2). To assist forming a monolayer of PDECs, the organoids were hand-picked manually and transferred to a fresh culture dish or trans-well membrane following breaking down of the Matrigel. During hand-picking, the organoids were separated from the Matrigel and collapsed by pipetting. Collapsed organoids attached to the surface, and PDECs started migrating out from organoids forming a monolayer (Supplementary Movie 1). From the monolayer of PDECs, single cells were harvested and embedded into fresh Matrigel. The PDECs re-formed organoid structures and grew over time (Fig. 1j). Furthermore, we have succeeded in freezing and reviving patient-derived PDECs using transformation of the organoid-monolayer structure (Fig. 1k). Hence, we have standardized the protocol to isolate, culture, and expand patient-derived ductal epithelial cells and pancreatic islets to provide a platform for the development of personalized medicine in pancreas-related disorders such as CFRD.

**Characterization of PDECs.** PDECs are one of the most abundant cell type present in the pancreas. We intended to confirm whether the isolated cells from the pancreatic remnant cell pellet were indeed PDECs. Isolated pancreatic ductal organoids (Fig. 2a) were first examined using standard morphological H&E staining (Fig. 2d). The images demonstrated that ductal epithelial cells are located at the edge of the organoid, with a central luminal area. Immunofluorescence images showed pancreatic ductal organoids expressing epithelial cell biomarkers, cytokeratin 19 (KRT 19) (Fig. 2b), E-cadherin (Fig. 2c), sodium transport channel (ENaC) (Fig. 2e), and tight junction protein (ZO-1) (Fig. 2f). We cultured PDEC-derived monolayer (Fig. 2g) from the ductal organoids and detected positive immunofluorescent signals corresponding to epithelial cell biomarkers, ZO-1 (Fig. 2h), F-actin, and KRT 19 (Fig. 2i). Paraffin-sectioned H&E images demonstrated a monolayer of PDECs (Fig. 2j) obtained from organoids. From the PDECs on the trans-well membrane, we observed CFTR located

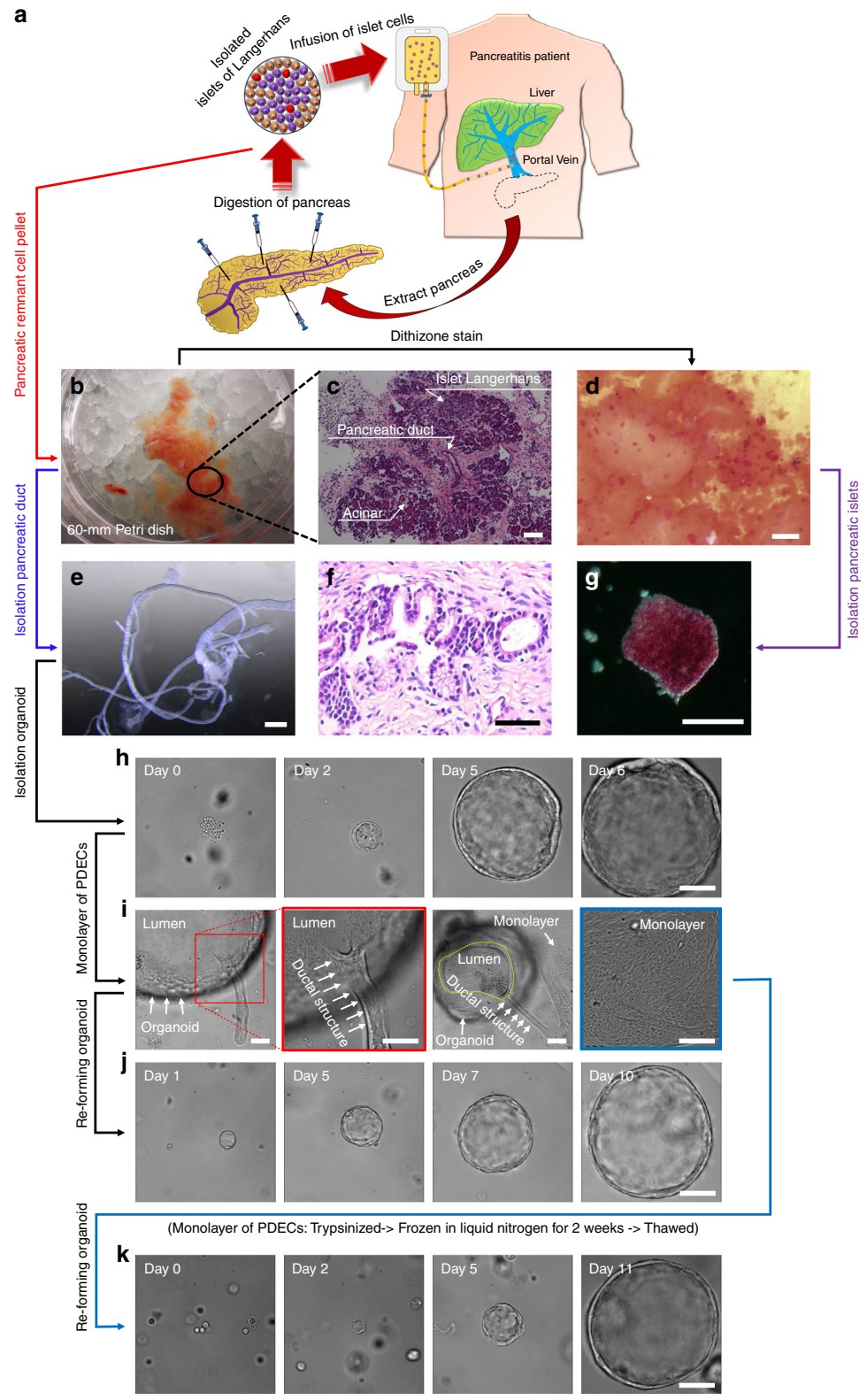

on the apical membrane of the ductal epithelial cells (Fig. 2k). Polarized monolayer of PDECs on a porous membrane in a pancreas-on-a-chip was examined with immunofluorescence image of ZO-1 and measurement of transepithelial electrical resistance using epithelial volt-ohm meter (Supplementary Fig. 3).

We performed RNA-sequencing in these organoids and verified that the organoids were of human PDECs in origin (Fig. 2l). Epithelial cell markers, cytokeratin family proteins (KRT 7, KRT 8, and KRT 19), CFTR, and E-cadherin were highly expressed in the organoids; however, blood cell marker (CDH 5), pancreatic acinar cell biomarkers, CPA1, GP2, and amylase (AMY2A), and pancreatic endocrine markers, insulin, glucagon, and somatostatin were not expressed. Hence, we sufficiently validated that the cell population processed from the pancreatic remnant cell pellet is purely ductal epithelium in origin, which is critical to developing our proposed model system to investigate

**Fig. 1** Isolation of patient-derived pancreatic ductal epithelium and islet cells. **a** Schematic representation of the total pancreatectomy with islet autotransplantation (TPIAT) procedure for pancreatitis patient. **b** Digested pancreatic remnant cell pellets were obtained following isolation of islet cells for infusion. **c** Hematoxylin and eosin (H&E stain) image demonstrates that the remnant cell pellet contains pancreatic islets, ductal epithelial cells, and acinar cells. **d** Pancreatic islets were identified by adding dithizone solution, where the color turned to red, and **g** isolated by manual pipetting. **e** Pancreatic ductal tissues were isolated by microdissection from the pancreatic remnant cell pellet following TPIAT. **f** H&E staining showed that pancreatic ductal epithelial cells were surrounded by collagen and connective tissue. **h** Pancreatic ductal epithelial cells (PDECs) were isolated from the ductal tissue and embedded in Matrigel matrix. PDECs grew into large spheres over time. **i** PDECs extend from the isolated organoid to form a monolayer on the surface of the substrate. **j** Ductal epithelial cell monolayers were re-formed into an organoid structure in the Matrigel and grew into a large sphere again. **k** Revived ductal epithelial cells following cryopreservation were embedded in the Matrigel and formed into large spheres over time. Scale bars: 50 μm (**f**), 100 μm (**c**, **g–j** and **k**), 500 μm (**e**), and 1000 μm (**d**)

the functional coupling of two specific cell types: PDECs and pancreatic islets.

**Functional measurements in PDECs and pancreatic islets.** PDECs) are reported to have the highest expression of CFTR in the body[1,11–13]. CFTR function in the pancreas has a critical role in maintaining fluid and pH within the pancreatic duct to deliver digestive enzymes secreted by acinar cells into the duodenum that are important for digestive function in the intestine. CFTR function was monitored in pancreatic ductal organoids in response to the cAMP-activating agonist forskolin (FSK; 10 μM). In this assay, CFTR function was reported as a measure of fluid secretion calculated by the ratio of luminal volume to that of the entire organoid[26]. During treatment with FSK, CFTR channels open, and chloride ions are pumped into the lumen creating an osmotic driving force for water to follow. Thus, fluid secretion is increased resulting in expansion of luminal volume. Fluid secretion was compared before and after treatment with FSK for 2 h, as shown in Fig. 3a. Basal secretion of the ductal organoids was 60% before treatment and increased to 75% upon incubation with FSK. The graph was obtained using over 450 organoids derived from 21 pancreatitis patients. Alongside, we cultured ductal organoid-derived PDECs on a trans-well membrane and monitored CFTR function using short-circuit current ($I_{sc}$) measurement (Fig. 3b). The trans-electrical resistance was 1200 Ω/cm² ($1.3 \times 10^5$ of cells). By activating the CFTR channel with FSK, electrogenic movement of chloride ions from the apical side of these cells generated $I_{sc}$ peak ($\Delta I_{sc} = 30$ μA/cm²) (Fig. 3b). The addition of CFTR inhibitor, CFTR$_{inh-172}$ (20 μM), caused dramatic decrease in the $I_{sc}$ (Fig. 3b). However, the context of CFTR function in the regulation of endocrine function by pancreatic islets remains to be elucidated. But before we investigated this possibility, we intended to monitor endocrine function in vitro. Pancreatic islets were isolated from the pancreatic remnant cell pellet of the same patient source as for PDECs and cultured in vitro using the methodology as described above (Fig. 3c). Pancreatic endocrine function consists of production of insulin (β cells) and glucagon (α cells) from pancreatic islets to maintain an appropriate blood glucose level. The pancreatic islets were examined by immunofluorescence staining specific to insulin and glucagon (Fig. 3d). We observed arrangement of β cells and α cells in the pancreatic islet marked by insulin (green) and glucagon (red), respectively (Fig. 3d). α Cells are located at the edge of the clustered pancreatic islets, while β cells were distributed uniformly across the islet (Fig. 3d). We successfully monitored endocrine function in the pancreatic islets by measuring the concentration of insulin in the culture media in response to variable concentrations of glucose, 100 mg/dL (equivalent to normoglycemia) and 450 mg/dL (equivalent to hyperglycemia) (Fig. 3e). We observed that pancreatic islets secreted significantly more insulin when exposed to high glucose conditions; from 23.4 μLU/mL insulin in low glucose-containing media upon incubation for 1 h (3.3 μLU/mL at time = 0 h) to 127 μLU/mL

insulin in high-glucose-containing media during another 1 h incubation (Fig. 3e). In order to verify that endocrine function in islet cells obtained from pancreatitis patient is not impaired, we compared insulin secretion response to highly concentrated glucose (450 mg/dL) from non-pancreatic patient and pancreatitis patient (Supplementary Fig. 4). Increased insulin secretion upon exposure to high-glucose-containing media was observed in both the patients. Thus, the overall function in islet cells was not impaired in pancreatitis patient.

Therefore, we generated robust in vitro functional systems to monitor CFTR function from PDECs and endocrine function from pancreatic islets to set forth the stage to study CFRD.

**A highly sensitive microfluidic device.** Using human tissue has its limitations, including limited availability and a very low viable cellular yield. The short-circuit current ($I_{sc}$) assay is the gold-standard method to monitor CFTR function in real time; however, it requires approximately $1.3 \times 10^5$ cells, and takes approximately 2 weeks to achieve a fully covered-polarized monolayer of epithelial cells on the trans-well membrane (33 mm²). Here, we developed a highly sensitive microfluidic device to monitor CFTR function from PDECs and insulin secretion from pancreatic islets cultured on the chip as shown in Fig. 4. The device, a single-channel chip (Fig. 4b), was designed to mimic pancreatic duct-like structure, which has branches with narrowing diameters (Supplementary Fig. 5). The chip was fabricated using standard photolithography and soft lithography, having dimensions 26.87 mm² (area), 0.14 mm (thickness), and 3.76 mm³ (volume) for cell culture (Supplementary Fig. 6). We cultured PDECs (Fig. 4a) and pancreatic islets (Fig. 4c) to monitor CFTR function and insulin secretion, respectively. Total amount of cell culture media needed in the chip is only 56 μL, which includes 3.76 μL for the cell culture chamber and 52 μL for two side tubings. This is in contrast to the required 200 μL (apical side) and 500 μL (basolateral side) for an Ussing chamber. We have successfully monitored CFTR function in PDECs with <10,000 cells using the iodide efflux assay 3 days after seeding cells (Fig. 4d). In the first step of iodide efflux, cells were loaded with the iodide. Upon CFTR activation using FSK at time-point 10 min following a baseline, the iodide was pumped out of the cells through CFTR channel, which gave an iodide peak ($60 \pm 18$ nM/μL).

Using the single-channel chip, we were able to detect insulin secretion in pancreatic islets (Fig. 4e). Pancreatic islets were cultured on the chip with 100 mg/dL glucose-containing media (basal medium). We obtained 3 μLU/mL insulin from 15 pancreatic islets in the basal medium wash with no incubation. The concentration of secreted insulin increased from 27 μLU/mL (1 h in the basal medium) to 106 μLU/mL (1 h in the high-glucose-containing medium) (Fig. 4e). In this manner, we developed a highly sensitive microfluidic device to measure CFTR function in PDECs and insulin secretion in islets with small numbers of cells.

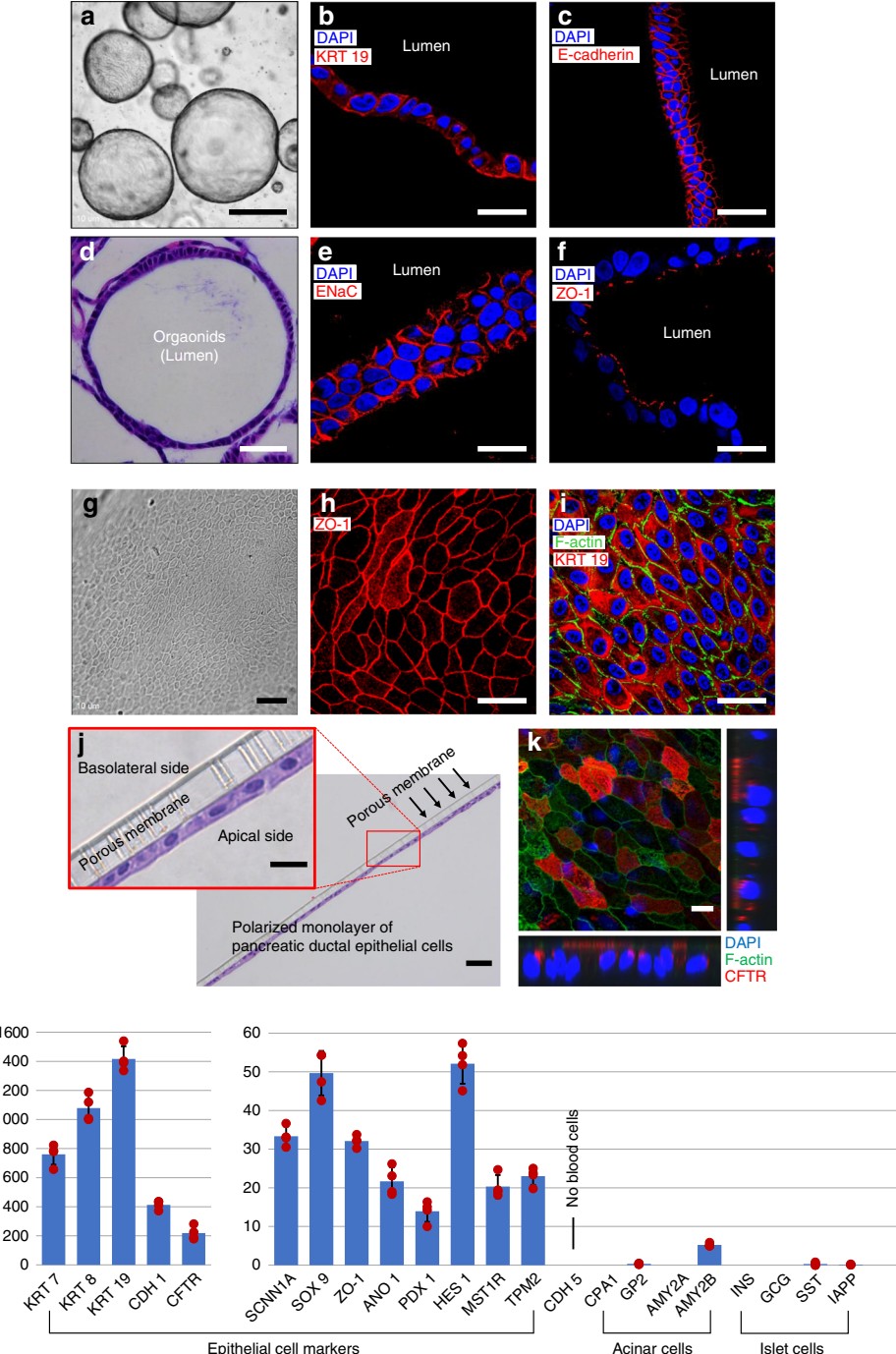

**Fig. 2** Characterization of pancreatic ductal epithelial cells. (**a–f**, **l**: Characterization of organoids). **a** Characterization of pancreatic ductal organoids using epithelial cell markers, **b** cytokeratin 19 (KRT 19, **c** E-cadherin, **e** sodium transport channel (ENaC), and **f** ZO-1. **d** Hematoxylin and eosin (H&E) image shows the orientation of pancreatic ductal epithelial cells in spheroid of the organoid. (**g–k**: Characterization of monolayer of pancreatic ductal epithelial cells (PDECs)). **g** Phase contrast and **j** H&E images show monolayer of PDECs formed from the organoids. Monolayer of PDECs showed positivity for tight junction **h** ZO-1, **i** F-actin and KRT 19, and **k** cystic fibrosis transmembrane conductance regulator (CFTR). **l** RNA-sequencing data was obtained from the pancreatic ductal organoids and verified the PDEC origin ($n = 4$ sample preparation from the same patient). Data are mean ± SD. Scale bars: 10 μm (**k**), 20 μm (**b**, **c**, **e**, **f**, **h–j** enlarge), 50 μm (**d**, **g**, **j**), and 500 μm (**a**)

**Pancreas-on-a-chip to study CF-related disorders**. We could detect that there is an interface between the ductal cells and islets based on H&E staining performed in a small piece (1 cm²) of non-treated tissue isolated from the head of the pancreas of a TPIAT patient (Fig. 5c) and immunostaining data in the same region, which was obtained from serial sections of the same sample, that showed CFTR-expressing ductal cells located in close proximity to insulin-expressing islets (Fig. 5a). Importantly, CFTR is only expressed in the PDECs, not in the pancreatic islets[12] (Supplementary Fig. 7). Given the cellular proximity between ductal cells and islets, we hypothesized that there is a functional coupling between these two cell types. To test this possibility, we developed a pancreas-on-a-chip involving co-culturing of ductal epithelial cells and islets in two-cell culture

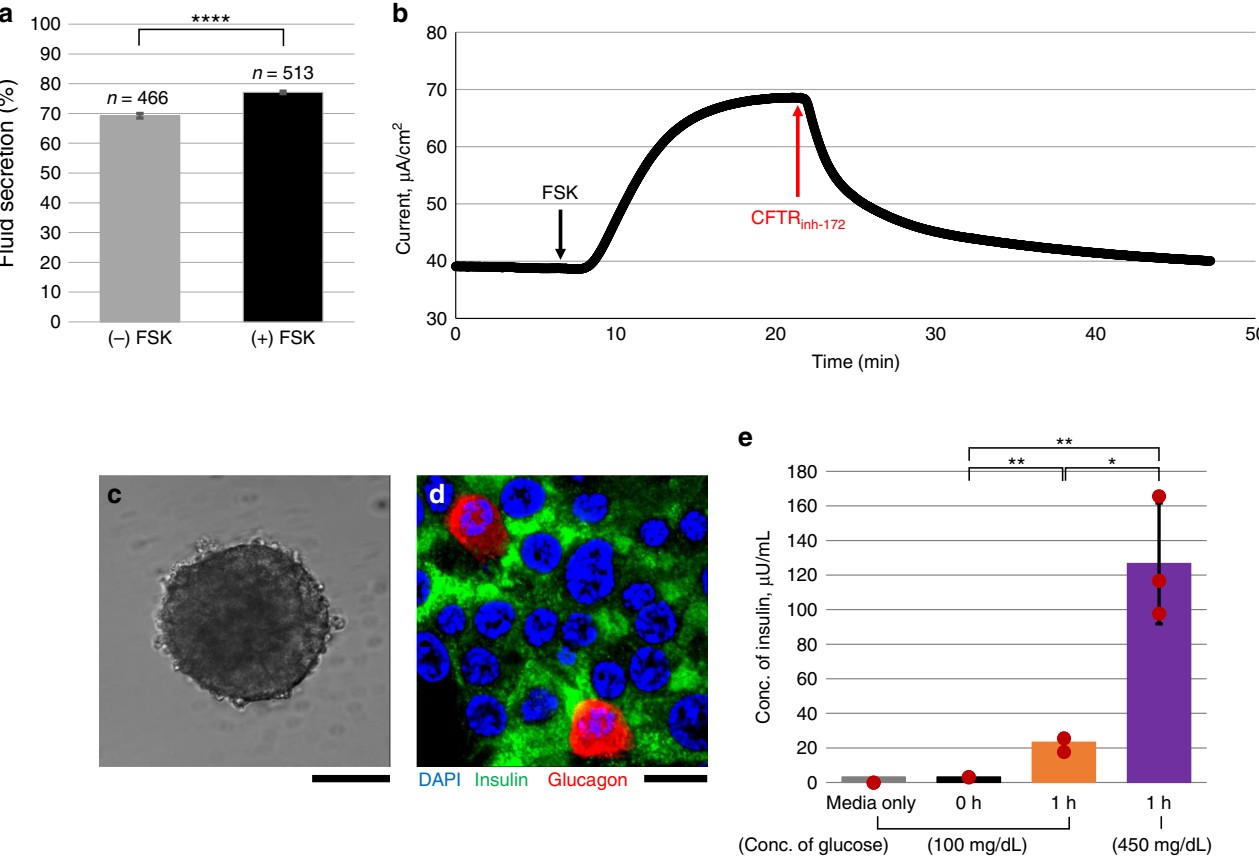

**Fig. 3** Monitoring cystic fibrosis transmembrane conductance regulator (CFTR) channel function and endocrine function. CFTR channel function was monitored by stimulating cAMP with forskolin (FSK) using **a** fluid secretion assay for pancreatic ductal organoids ($n > 450$ organoids; from 21 pancreatitis patients; data are mean ± SE) and **b** short-circuit current measurement in polarized monolayer of ductal epithelial cells grown on a trans-well filter. CFTR channel is activated by FSK and inhibited by CFTR$_{inh-172}$. **c** Phase contrast image shows cultured pancreatic islet in vitro. **d** Pancreatic islets were examined by immunofluorescence detection of insulin (green) and glucagon (red). **e** Endocrine function was monitored by incubating pancreatic islets with different concentrations of glucose-containing media (100 and 450 mg/dL) for 1 h serially. Pancreatic islets were stimulated by high glucose ($n = 3$ sample preparation from the same patient; data are mean ± SD). Scale bars: 100 μm (**c**) and 20 μm (**d**). ($p$ values from one-way analysis of variance (ANOVA) and adjust using Bonferroni factor: *<0.01, **<0.005, ****<$1.0 \times 10^{-20}$)

chambers separated by a thin layer of porous membrane (<10 μm thickness) (Fig. 5b and Supplementary Fig. 8). The PDECs were cultured on the top chamber and pancreatic islets were seeded in the bottom chamber (Fig. 5d). Next, we tested how CFTR function may affect insulin secretion from the islet cells in this double-channel chip system. We measured secreted insulin in 1-h increments from pancreatic islets in the bottom chamber following stimulation or inhibition of CFTR channel function (Fig. 5e). Stimulation of CFTR channels in PDECs in the top chamber did not show significant changes. On the other hand, upon inhibition of CFTR channel function in PDECs using CFTR$_{inh-172}$ (Chip B; 20 μM, 1 h), insulin secretion was significantly decreased (53.7%; Δ191.4 μLU/mL). We examined if the CFTR inhibitor directly affects insulin secretion from the islet cells (Supplementary Fig. 9). We cultured islet cells in the double-channel chip without PDECs (Supplementary Fig. 9a) or co-cultured with PDECs lacking pores on the membrane to perturb communication between PDECs and islet cells (Supplementary Fig. 9b) and added 20 μM CFTR$_{inh-172}$ to the top chamber. We observed that inhibition of CFTR under these conditions did not influence insulin secretion. Insulin secretion from the islet cells was not altered in the presence of FSK (10 μM) and CFTR$_{inh-172}$ (20 μM) (Supplementary Fig. 10). FSK stimulation did not significantly alter insulin secretion from the islet cells maintained in the basal medium (100 mg/dL glucose). At the end

of the experiment, islet cells were directly exposed to high-glucose-containing media (450 mg/dL glucose; Chip A and Chip B) to verify its responsiveness to the glucose challenge, suggesting that the endocrine function was not impaired. Insulin secretion increased in both chips (Chip A: Δ60.7 μLU/mL and Chip B: Δ181.4 μLU/mL) and the amount of insulin secreted was higher than before stimulation or inhibition of CFTR function in PDECs.

To further consolidate the possibility that CFTR function directly affects insulin secretion, we examined the cell–cell functional correlation between PDECs and islet cells derived from pancreatitis/CF patient, who was diagnosed with very mild CF and underwent TPIAT (Supplementary Table 1). The patient has ΔF508 (allele 1), R117H (allele 2), and heterozygote for SPINK1 mutation. This patient was diagnosed to have mild CF and has some CFTR function as demonstrated by the mild phenotype (i.e., body mass index: 19.84; sweat chloride: 51 mmol/L; forced expiratory volume in 1 s predicted: 114% and is not diabetic). Additionally, we monitored CFTR function using fluid secretion assay and endocrine function using enzyme-linked immunosorbent assay (ELISA) as described earlier prior to co-culture of the two cell types in pancreas-on-a-chip. We observed that the pancreatic ductal organoids showed partially impaired CFTR function (20% lower than non-CF pancreatitis patient in basal secretion and under 5.3% in FSK-stimulated secretion). Islet

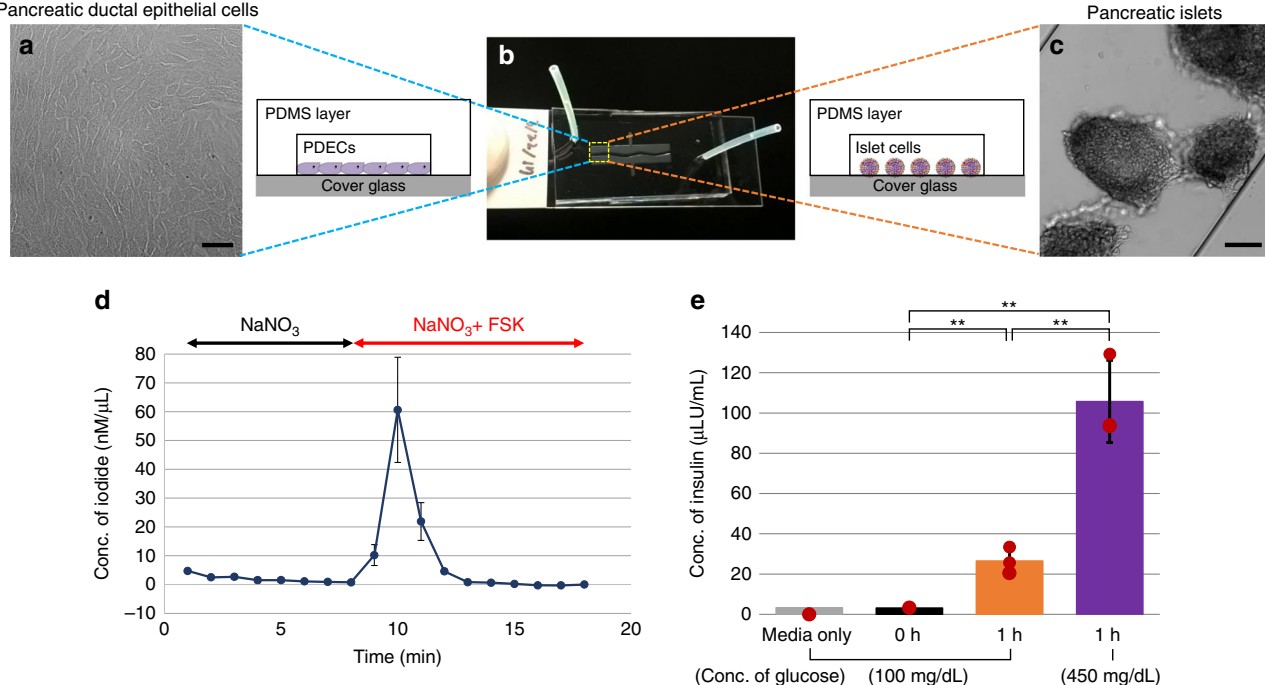

**Fig. 4** A unique microfluidic device. **b** A microfluidic device, single-channel chip, was designed to mimic pancreatic duct-like structure with branches and narrowing diameters. **a**) Pancreatic ductal epithelial cells (PDECs) were cultured in the chip and (**d**) cystic fibrosis transmembrane conductance regulator (CFTR) function was monitored using iodide efflux assay with <10,000 PDECs ($n = 3$). **e** Endocrine function was monitored with 15 pancreatic islets ($n = 3$) **c** by incubating with 100 and 450 mg/dL glucose-containing media for 1 h serially. Secreted amount of insulin was measured using enzyme-linked immunosorbent assay (ELISA). Scale bars: 50 μm (**a**) and 100 μm (**c**). (p values from one-way analysis of variance (ANOVA) and adjust using Bonferroni factor: **<0.005; $n = 3$ number of chips; data are mean ± SD)

cells secreted insulin in response to the glucose challenge (Supplementary Fig. 10a, b). We co-cultured PDECs and islet cells in pancreas-on-a-chip and measured insulin secretion from the islet cells as described earlier. We observed similar trend that inhibition of CFTR function affected endocrine function. Insulin secretion was decreased in pancreatitis/CF patient-derived pancreas-on-a-chip by 7.9%, but it was not significant (Supplementary Fig. 10c, d).

Overall, using this unique pancreas-on-a-chip device, we demonstrated that ductal cells and islets are functionally coupled, a first-of-a-kind observation that CFTR plays a role in directly regulating insulin secretion. This observation is directly relevant to CFRD in which there is a loss of CFTR function.

## Discussion

We have successfully isolated patient-derived pancreatic ductal organoids following TPIAT, and we have generated a freezing and reviving protocol for pancreatic ductal epithelial cells. Pancreatic ductal organoids demonstrated growth into large spheres over time. The organoids cultured in 3D matrix positions us to efficiently harvest pure pancreatic ductal epithelial cells among multiple cell types that are present in the pancreatic remnant cell pellet. The organoids were grown effectively from a limited number of cells to form a functional unit. The 3D organoid formation with luminal area internally has been observed in other organs, including lung[27], liver[28], and intestine[29]. However, this finding is our repeated observation of duct-like formation from the pancreatic ductal organoids. While the mechanism is currently unclear, further investigation of this ductal formation may elucidate mechanisms involved in the development of the pancreatic duct in vivo.

Pancreas-on-a-chip mimics in situ pancreatic cell function and interface compared to conventional human cell culture model.

The chip allows to mimic fluid flow in vivo by setting a perfusion system in a cell culture incubator or on a microscope, relevant mechanical cues in cellular signaling, and allows tissue–tissue interface (i.e., duct-islet) to study cell–cell signaling[30]. Pancreas-on-a-chip helps answer the fundamental question in CFRD: is loss of CFTR function in PDECs primary to CFRD development. Based on our data, it is indeed the case. Surprisingly, the absolute amount of insulin was around 50% decreased during inhibition of CFTR channel function. In the human pancreas, the organ system is extremely complex in physiological and pathological perspectives. However, this finding tells us that CFTR channel function plays a critical role in maintaining endocrine function and may provide critical insight into the etiology of CFRD. To investigate the cross-talk between PDECs and pancreatic islets, metabolism studies of these two cell types may need to be performed.

CFRD is a serious complication in CF patients who in general have disordered glucose metabolism with increasing risk with advancing age. Using this in vitro chip model, we can study CFRD and glucose imbalance in CF individuals, assay variability in the glucose measures in these individuals, determine correlation of glucose levels with the CFTR mutation type, and test small-molecule interventions (i.e., approved CFTR modulators) that may improve glucose abnormalities in the patient samples. Our data based on the effect of CFTR-specific inhibitor and lack of function mutation in CFTR strongly suggests that CFTR function modulates insulin secretion that underlies the pathology of CFRD.

This patient-derived in vitro model system also allows the development of personalized medicine with highly sensitive measurements of epithelial and/or endocrine functions from the pancreatic cells. Because the cells cultured in the chip are all patient derived, we can easily and quickly obtain other clinically relevant measures using this model in safe manner. Alcohol abuse

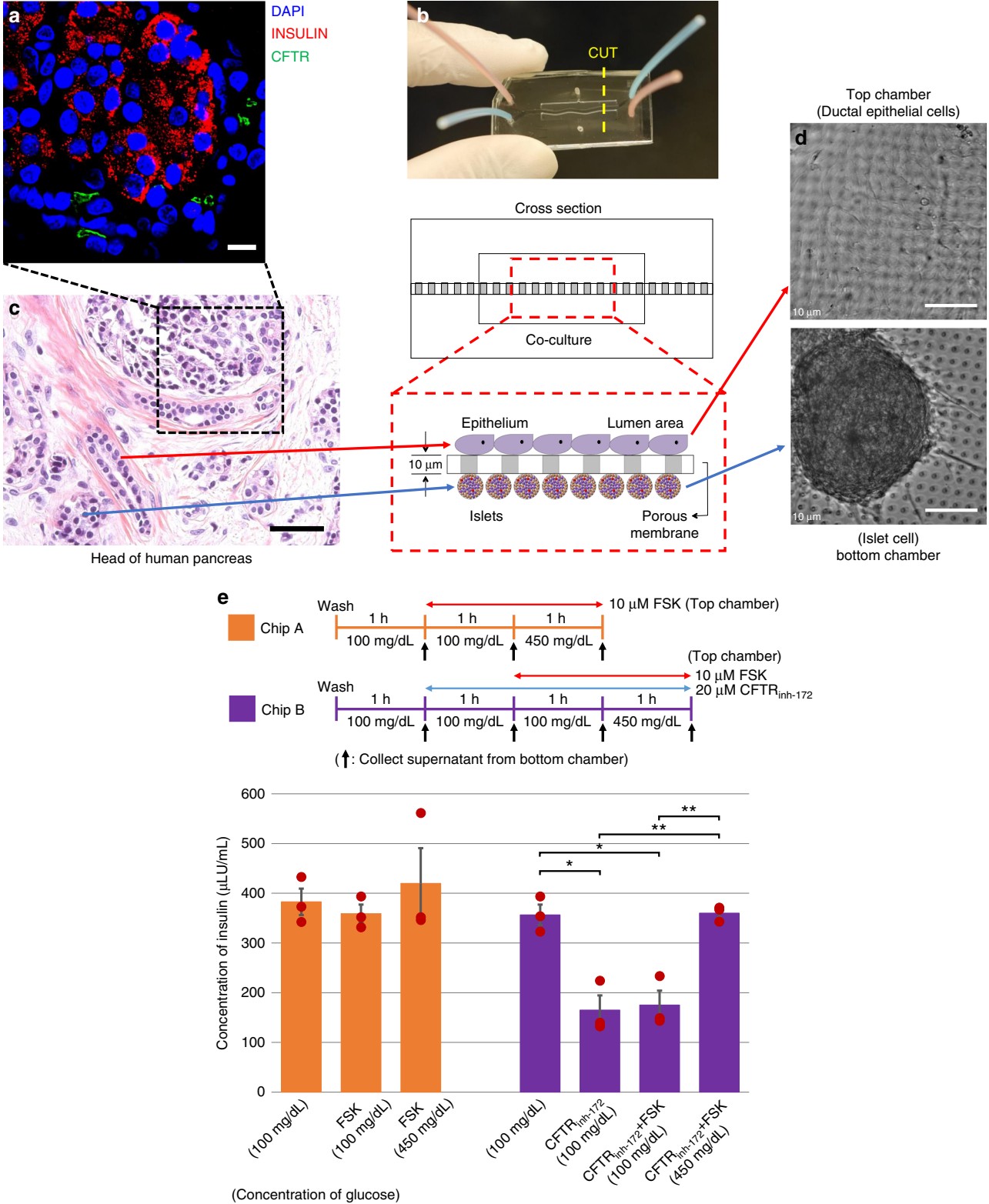

**Fig. 5** Pancreas-on-a-chip to study cystic fibrosis-related diabetes (CFRD). A small piece of non-treated head of pancreas was obtained and examined by **a** immunofluorescence microscopy with insulin and cystic fibrosis transmembrane conductance regulator (CFTR) and **c** hematoxylin and eosin (H&E) stain. It shows that pancreatic islets are located in close proximity to the pancreatic duct. To mimic pancreatic structure and function, **b** pancreas-on-a-chip has been developed that is comprised of two-cell culture chambers and a thin layer of porous membrane. **d** Pancreas-on-a-chip allows us to co-culture pancreatic ductal epithelial cells (PDECs) on the top chamber with pancreatic islets in the bottom chamber. **e** Endocrine function of islet cells was monitored with stimulation or inhibition of CFTR function in PDECs on the top chamber. We observed that CFTR channel function has a direct effect on the endocrine function. Secreted insulin was dramatically decreased (53.7%) by inhibition of CFTR function of PDECs. This in vitro model system, pancreas-on-a-chip, allows us to study cell–cell interaction. Scale bars: 10 μm (**a**), 50 μm (**c**), and 100 μm (**d**). (p values from one-way analysis of variance (ANOVA) and adjust using Bonferroni factor: *<0.05, **<0.005; number of chips: Chip A (n = 3) and Chip B (n = 4); data are mean ± SE)

has been reported to lead to dysfunction and degradation of CFTR protein on the apical membrane of the epithelium[31]. Using this chip model, we can monitor CFTR function and/or endocrine function in response to alcohol in a dose-dependent manner that is not possible in patients.

The microfluidic device can be set up for multiple analyses, including functional assays and microscopic measurements in real time. This in vitro model system will facilitate drug discoveries. However, the polydimethylsiloxane (PDMS) used for the cell culture chambers has a challenging property, which is that the hydrophobic PDMS absorbs hydrophobic small molecules[32,33]. After oxygen plasma treatment, it changes to highly hydrophilic[34]. However, it recovers to hydrophobic over time[35]. The hydrophobic surface interferes with cell adhesion on the substrate[36]. Alternatively, other materials for fabricating the microfluidic device have been adopted, such as poly methyl methacrylate[37], acrylonitrile butadiene styrene copolymer[33], cyclic olefin copolymer[38], and styrene ethylene butylene styrene[39]. However, those materials also have limitations when mimicking human organ systems due to their rigidity and brittleness, leading to a difficult fabrication process. The Kang laboratory has demonstrated that PDMS-based microfluidic devices can be maintained as a hydrophilic surface for weeks after treatment with oxygen plasma[34]. Other groups showed that sol-gel-modified PDMS[40] and bovine serum albumin-coated -PDMS[41] can minimize absorbance of hydrophobic drugs by PDMS. Alternatively, collagen coating of the chamber can be utilized to increase cell adhesion, as we used in our model system. Although these limitations are yet to be overcome, PDMS is still the best material to mimic organ systems.

In summary, we have isolated and cultured patient-derived pancreatic cells, PDECs, and pancreatic islets from the same patient. This efficient and highly reproducible method allows us to widely study relevant pancreatic disorders. Moreover, the in vitro model system, pancreas-on-a-chip, helps us to investigate the cross-talk between PDECs and islet cells in the development of disease pathologically and physiologically. This pancreas-on-a-chip model system, with its highly sensitive profile, can allow for early diagnosis and individual diagnosis that may help prevent or reduce the progression of disorders such as CFRD, and additionally, can afford the opportunity for drug discovery and personalized medicine in such disorders.

## Methods

**Human studies**. Human tissue, pancreatic remnant cell pellets were collected according to standard research protocols approved by the Institutional Review Board and Department of Pathology at Cincinnati Children's Hospital (IRB: 2014-6279; renewed 27/11/2017).

**Cell culture media**. For PDECs, advanced Dulbecco's modified Eagle's medium/nutrient mixture F12 (DMEM/F12) (Invitrogen; #12634010) with 10 mM HEPES (Invitrogen; #15630-080), GlutaMAX (1×; Invitrogen; #35050-061), and penicillin streptomycin (PS) (1×; Invitrogen; #15140-122) was used as base organoid media (I). Organoid media (II) contains N2 (1×; Invitrogen; #17502-048), B27 (1×; Invitrogen; #17504-044), and 1 mM N-acetylcysteine (Sigma; #A7250-100G) in organoid media (I). Organoid media (III) contains growth factors, supplements 100 ng/mL epidermal growth factor (R&D System; #236-EG-200), 50 ng/mL R-spondin (R&D System; #4645-RS-025/CF), and 100 ng/mL Noggin (R&D System; #6057-NG-025) in organoid media (II). ROCK inhibitor, 10 μM Y-27632 (BD Biosciences; #562822), was added for the first 4 days followed by isolation of pancreatic ductal epithelial cells.

Islet cells were cultured in RPMI-1640 (Invitrogen; #61870036) containing PS (1×), 10% fetal bovine serum (FBS) (Atlanta Biologicals; #S11150 premium), and 10 μM Y-27632 for the first day of isolation. The RPMI-1640 media were switched to low glucose-containing DMEM (Invitrogen; #11885-084; 100 mg/dL glucose) with PS (1×) and 10% FBS from the second day of isolation. High-glucose-containing DMEM (Invitrogen; #11960-044; 450 mg/dL glucose) was used to stimulate pancreatic islets.

**Isolation of pancreatic ductal organoids and islet**. Pediatric patients with severe acute recurrent or chronic pancreatitis undergo TPIAT. During the TPIAT, the excised pancreas was surgically dissected and digested to isolate pancreatic islets for infusion into the liver through the portal vein. We obtained discarded pancreatic remnant cell pellets following isolation of pancreatic islets. The pancreatic remnant cell pellet still contains pancreatic islets, ductal epithelial cells, and acinar cells (Fig. 1c). From the remnant cell pellet, we isolated pancreatic ductal tissues by microscopic dissection under a stereo microscope (Leica; #M165FC) (Fig. 1e). The pancreatic remnant cell pellet was prepared on a 60-mm dish containing phosphate-buffered saline (PBS) by dissecting a white cluster (Fig. 1b; circle, Fig. 1d) using two forceps until the appearance of pancreatic duct-like structure remained. Then, holding the cluster using one forcep the pancreatic duct is gently pulled out using the other forceps. The pancreatic duct is extracted smoothly, because surrounding connecting tissues were uniformly oriented along the length of the duct. Microdissection scissors were used to remove other cell types attached to the ductal tissues, if necessary. Isolated ductal tissues were treated with 2 mM EDTA (Invitrogen; #15575) in 10 mL PBS on a shaker (Nutator; #421105) at 4 °C for 40 min. The tissues were filtered through a 70 μm strainer (Falcon; #352350) to remove the EDTA solution, and all tissues were transferred to fresh 15 mL tube containing 10 mL PBS. The tube was shaken mechanically to help separate ductal epithelial cells apart from the tissues. The supernatant was then filtered through a fresh 70 μm strainer and 10% FBS was added to stop the proteolytic activity. Supernatant was discarded after centrifugation (Beckman Coulter; #Allegra X-14R) at 233 × g for 3 min. Organoid media (II) were added and the ductal epithelial cells were re-suspended by pipetting gently. Matrigel (Corning; #356231) was added at a ratio of 2:3 vol% growth media to Matrigel and mixed well avoiding any bubbles. Matrigel (50 μL) was plated with ductal cells on a substrate and incubated at 37 °C for 15 min for solidification of the Matrigel. The Matrigel was covered with 500 μL organoid media (III) containing growth factors. For the first 4 days, 10 μM Y-27632 of ROCK inhibitor was added to help cellular recovery from cellular aggregates[42,43]. Organoid media (III) were refreshed every other day.

Pancreatic remnant cell pellets contain leftover islet cells after isolation of islets for infusion. Islet cells can be easily identified by dithizone staining (100 μg/mL), because dithizone binds to the zinc ion presented in insulin secreted by β cells in the pancreatic islets[25]. Dithizone solution is green, but it turns to red when it binds to the zinc ion (Fig. 1d, g). Islet cells were transferred gently to a fresh plate using a 200 μL pipette and cultured in RPMI-1640 media containing 10 μM Y-27632 for the first day. Next day, the media were switched to low glucose-containing DMEM (100 mg/dL) and refreshed every other day.

**Obtaining monolayer of pancreatic ductal epithelial cells**. Pancreatic ductal organoids were grown over time in Matrigel (Fig. 1h). The mechanism is unknown, but we continuously observed that when the ductal organoids contact the surface, they start to form duct-like structures and then a complete monolayer (Supplementary Fig. 2). To assist forming a monolayer of PDECs, the Matrigel was broken down by pipetting with 1 mL organoid media (I) when the average diameter of the organoids reached 500 μm. Organoids were then transferred to 1.5 mL tube by using a 200 μL pipette. After centrifugation at 8600 × g for 3 min (Eppendorf microcentrifuge; #5418), the supernatant was discarded and the organoid media (III) containing 10 μM Y-27632 were added to the pellet. Organoids were then plated on a fresh dish or trans-well membrane. Care must be taken to separate the Matrigel completely from the organoids, or they will not adhere, and will not survive.

**Freezing and reviving pancreatic ductal epithelial cells**. To cryopreserve PDECs, PDEC monolayers were trypsinized with 0.5% trypsin EDTA (1×; Invitrogen; #15400-054) at 37 °C for 10 min to detach cells after washing cells with PBS and transferred to 15 mL tube containing 5 mL organoid media (I) with 10% FBS and 10 μM Y-27632. Cell pellets were obtained after centrifuging at 233 × g for 5 min. The supernatant was discarded and cells were re-suspended with freezing media (Invitrogen; #12648010) containing 10 μM Y-27632. Epithelial cells were transferred to a cryopreservation tube and placed on dry ice immediately and stored at −80 °C. For long-term storage, the cells were stored in liquid nitrogen.

To revive PDECs, the cells were thawed quickly at 37 °C and all supernatants were transferred to 15 mL tube containing 5 mL organoid media (I) with 10 μM Y-27632. After centrifugation at 233 × g for 5 min, the supernatant was discarded. Appropriate organoid media (II) with Matrigel were added and 50 μL Matrigel was plated with cells on plates to form organoid structure as before. The Matrigel was covered with organoid media (III) containing 10 μM Y-27632 after incubation at 37 °C for 15 min.

**Fabrication of pancreas-on-a-chip**. Our customized microfluidic device was designed to mimic ductal structure having branches with narrowing diameters (Supplementary Fig. 5). The design was drawn using the AutoCAD software. The chip was fabricated through the standard photolithography and soft lithography techniques (Supplementary Fig. 6). Initially, the silicon wafer was washed with acetone, isopropanol (IPA) and water. It was placed on a hot plate at 60 °C for 10 min to dry thoroughly after air drying. After cooling down to room temperature, a negative photoresist SU-8 (Microchem; #Y131269) was applied to the wafer using

a spin coater (Specialty Coating Systems; #6800) by the following process: (1) Place the wafer on the vacuum chuck of the spin coater and drop appropriate SU-8 on the wafer. (2) Ramp up to 500 rpm for 10 s and hold for 10 s. (3) Increase the speed to 1000 rpm for 10 s and hold it for 15 s for 140 μm thickness of cell culturing chamber in the chip. (4) Speed down to 0 rpm for 10 s. The wafer was placed on the hot plate and baked at 65 °C for 10 min and at 95 °C for 30 min serially. The wafer was exposed to ultraviolet (UV) light (wavelength: 365 nm; exposure energy: 240mJ/cm$^2$) through a patterned photomask for 20 s after cooling down to room temperature. The wafer was baked on the hot plate at 65°C for 1 min and at 95 °C for 20 min and was cooled down to room temperature. The wafer was immersed into SU-8 developer (Fisher Scientific; #NC9901158) for development process of unexposed area to UV light. After completion of development, the wafer was washed with IPA and dried with filtered air. The patterned silicon wafer was then baked on the hot plate at 150 °C for 30 min. After cooling down to room temperature, the patterned wafer can be used as a mold. These standard photolithography procedures were carried out in in a 100-class clean room.

For this microfluidic device, we used flexible, transparent, and low-cost materials, PDMS (Ells Worth Adhesive; #4019862). We mixed viscous PDMS with curing kit at the ratio of 10:1 (wt%) and degassed in a desiccator to remove bubbles. In the meantime, the patterned silicon wafer was treated with trichloro silane (Sigma-Aldrich; #448931) for 30 min in another desiccator to assist peeling off the patterned PDMS layer from the wafer. The uncured PDMS was cast onto the wafer and cured at 60 °C for at least 4 h. The solidified patterned PDMS layer was peeled off from the wafer and holes were created at both ends of the cell culture area for seeding and feeding cells. The patterned PDMS layer and a cover glass were treated with oxygen plasma for 30 s using Tergeo Plasma Cleaner (PIE Scientific) and immediately assembled together. It was placed on the hot plate at 120 °C for 30 min to seal completely that is a single-channel chip. The activated surface of the patterned PDMS layer and cover glass by plasma treatment becomes highly hydrophilic with polar characteristics[34]. This enhances the bonding process of the two surfaces.

Pancreas-on-a-chip is comprised of top and bottom layers for cell culture chambers and a thin layer of porous membrane to separate the two chambers as double-channel chip. Patterned PDMS layers of top and bottom chambers are prepared as described previously for single-channel chip. Holes were created through the PDMS layer of the top chamber for seeding and feeding cells before assembly with the porous membrane. For the thin layer of porous membrane, a mold was fabricated of uniformly arranged cylinders, with 10 μm diameters, 25-μm gaps, and 40-μm thickness, on a silicon wafer through the photolithography. The wafer is coated with trichloro silane in the desiccator for 30 min. In the meantime, RTV615 (Momentive; #9480), which shows large linear behavior of stain and promotes fabrication of a thin layer uniformly comparing to PDMS[44,45], is mixed with a curing kit at the ratio of 5:1 (wt%) and degassed in the desiccator for 30 min. The patterned wafer was placed on the spin coater and spun after covering the pattern with degassed RTV615 as the standard for 10 μm thickness of porous membrane; thus, (1) ramp up to 500 rpm for 10 s and hold for 10 s; (2) increase the speed to 3000 rpm for 10 s and hold it for 5 min; (3) speed down to 0 rpm for 10 s. Leave the wafer at room temperature for 10 min for uniform surface and incubate at 60 °C for 10 min for partial solidification of the surface. After incubation, the top PDMS layer was placed, patterned face down, directly onto the cylinders and slightly pressed onto the PDMS layer for contacting the surface of top layer to the partially cured RTV615. The top chamber is incubated with the porous membrane on the wafer overnight and cooled to room temperature. The top chamber with the porous membrane is peeled from the wafer and holes created through the porous membrane to connect to the bottom chamber only. The top chamber and bottom chamber are aligned after oxygen plasma treatment and placed on the hot plate at 120 °C for 30 min to seal the double-channel chip.

Before seeding cells, the cell culture chambers were sterilized with 70% EtOH for 10 min and washed with autoclaved water using a needle (BD Biosciences; #305175; 20 G) and syringe (BD Biosciences; #309657; 3 mL). The chambers were coated with 50 μg/mL collagen (Sigma-Aldrich; #C3867-1VL) for 1 h at 37 °C and washed with PBS to increase cell adhesion. The microfluidic device was connected to a peristaltic pump (Cole-Parmer; #ISMATEC Reglo ICC) with tubing (Cole-Parmer; #97619-09) and supplied organoid growth media (III) at the flow rate of 1 μL/min to feed cells continuously. To feed cells manually, a syringe and needle was used.

**Culture cells in the microfluidic device.** Monolayers of PDECs were treated with 0.5% Trypsin EDTA (1×) at 37 °C for 10 min and floating cells were transferred to a 15 mL tube containing 5 mL organoid media (I) with 10% FBS and 10 μM Y-27632. The supernatant was discarded after spinning down at 233 × g for 5 min (4 °C) and cells were re-suspended with 120 μL organoid media (III) containing 10 μM Y-27632. Cells were transferred (2 × 10$^5$ cells/mL; 10,000 cells/chip) in the cell culture chamber coated with collagen (50 μg/mL) using a syringe and needle through a tubing (5 cm length) inserted through the PDMS layer. After overnight incubation at 37 °C, 5% CO$_2$ media were refreshed.

Pancreatic islets in 24-well plate were washed with PBS and incubated with 200 μL 0.5% Trypsin EDTA (1×) at 37 °C for 3 min for trypsinization. Pancreatic islets were transferred to a 1.5 mL tube filled with culture media. The supernatant was discarded after centrifugation at 8600 × g (microcentrifuge) for 3 min and cells

were re-suspended with 120 μL media. Pancreatic islets were transferred (300 islets/mL; 15 islets/chip) into the cell culture chamber using a syringe and needle. Media were refreshed after the pancreatic islets attached onto the surface of the chip.

**Immunofluorescence microscopy.** Pancreatic ductal organoids in Matrigel were fixed with 3.7% formaldehyde for 15 min at room temperature and the Matrigel was broken down by pipetting with 1 mL EtOH. The organoids were embedded into HistoGel (Invitrogen; #HG-4000-012) and were first examined by gold-standard morphological section and H&E stain. Paraffin-sectioned organoids were deparaffinized for immunofluorescence microscopy. For a monolayer of PDECs on a trans-well membrane or a pancreas-on-a-chip, cells were fixed with 3.7% formaldehyde for 15 min at room temperature. Cells were then permeabilized using 1× permeabilization solution (eBioscience; #00-8333-56) for 8 min at room temperature and washed three times with PBS for 5 min each. Cells were then blocked using 1% goat serum (Sigma-Aldrich; #A8806-5G) for 1 h at room temperature and incubated with primary antibodies (diluted in antibody diluent (Invitrogen; #TA-125-ADQ) 1:100), anti-CFTR R1104 (Eric Sorscher lab, CF Center, University of Alabama, Birmingham, AL, USA [presently, Emory University, Atlanta, GA, USA]), anti-ZO-1 (BD Biosciences; #610967), anti-ENaC (Invitrogen; #PA1-920A), anti-KRT 19 (Invitrogen; #MA5-12663), anti-E cadherin (Cell Signaling Technology; #3195), anti-insulin (Cell Signaling; #C27C9), and anti-glucagon (Sigma; #G2654) overnight at 4 °C. Cells were washed three times with PBS for 5 min each and incubated with secondary antibodies (Invitrogen; Alexa Fluor 488 or 568; 1:500) for 1 h at room temperature. Alexa Fluor 488 Phalloidin (Invitrogen; #A12379; 1:50) was employed to the secondary antibody for F-actin staining following washing three times with PBS for 5 min each. Cells were incubated with DAPI solution (Invitrogen; #D1306; 1:500) for 20 min for nucleus staining and washed with PBS. For the trans-well membrane, cut edge of the membrane and transferred the membrane with cells onto a glass slide oriented cell-side up. Cells were then mounted in Vecta-shield mounting medium (Vector Labs; #H-1000). A cover slip was placed onto the cells and fixed with nail polish. For the pancreas-on-a-chip, cell culture chambers were separated manually by hands followed by nuclear staining with 4′,6-diamidino-2-phenylindole (DAPI) solution for 20 min. The porous membrane with cells remained on the upper layer. One drop of mounting solution was applied onto the cells and a coverslip was placed for imaging. Fluorescence images were obtained using a confocal microscope (Olympus FV1200). Combined images were created using an Image J software provided by NIH.

**Extract RNA from pancreatic ductal organoids.** Organoid growth media were discarded and the Matrigel was broken down by pipetting with 1 mL PBS. Pancreatic ductal organoids were picked and transferred to 1.5 mL RNA-free tube manually using a 200 μL pipette. The supernatant and Matrigel were discarded after microcentrifuge at 16,800 × g for 5 min and RNA was extracted using an Ambion miRNA Isolation Kit (Invitrogen; #AM1561) using the protocol provided by Ambion.

**Monitoring CFTR function.** CFTR function of pancreatic ductal organoids was monitored using the fluid secretion assay[26] in response to an intracellular cAMP-activating agonist (FSK; 10 μM) for 2 h at 37 °C. Fluid secretion was calculated by measuring the volume ratio of luminal area over the entire organoid pre-treatment and post treatment with FSK. Fluid secretions were monitored at day 4 after isolation of organoids with at least 20 organoids. The area of lumen and outer sphere was measured using the Image J software.

Pancreatic ductal organoids were transferred, when their diameter reached 500 μm, onto trans-well membranes (Corning; #3470), 10 organoids each as previously described. The ductal epithelial cells transformed into a polarized monolayer from spheroids on the trans-well membrane within 2 weeks. Transepithelial electrical resistance was measured using epithelial volt-ohm meter (World Precision Instruments, #EVOM and #STX2) and the trans-well membrane was mounted in an Ussing chamber when the resistance was over 1000 Ω/cm$^2$. Cells were bathed in Ringer's solution (mM) for apical side (pH 7.2): 0.12 NaCl, 25 NaHCO$_3$, 3.3 KH$_2$PO$_4$, 0.83 K$_2$HPO$_4$, 1.2 CaCl$_2$, 1.2 MgCl$_2$, 141 Na-gluconate, and 10 mannitol, and for basolateral side (pH 7.2): 120 NaCl, 25 NaHCO$_3$, 3.3 KH$_2$PO$_4$, 0.83 K$_2$HPO$_4$, 1.2 CaCl$_2$, 1.2 MgCl$_2$, and 10 D-glucose maintained the temperature of the bath using circulate system as 37 °C[26,46]. CFTR function was monitored in real time in response to current changing by FSK. When the current showed a stable baseline, 10 μM FSK was added to the apical side for CFTR channel opening. For CFTR channel closing, CFTR channel inhibitor, CFTR$_{inh-172}$ (20 μM), was applied to the apical side.

CFTR function of PDECs was monitored using iodide efflux assay[47]. Cell culture media were washed out with 136 mM NaNO$_3$ and incubated with 136 mM NaI for 1 h at 37 °C. After 1-h incubation, cells were washed with 300 μL of NaNO$_3$ (136 mM) and the supernatant was collected with 136 mM NaNO$_3$ using a syringe and needle. A 1.5 mL tube was placed on a digital weighing scale and the supernatant was dropped into the tube with recording the weight for approximately 20 μL of each sample. The first 10 samples were collected with 136 mM NaNO$_3$ and the other 10 samples with 136 mM NaNO$_3$ containing 10 μM FSK. Iodide concentration was calculated using an electrolyte detector (Thermo

Orion; #420) with electrode probe filled with specific iodide-sensitive electrolyte (Invitrogen; #900063). The electrode was immersed in 5 mL of 100 mM $NaNO_3$ (stirred) to detect iodide. Voltage change was measured by adding each sample serially. A standard curve was obtained using 10 μM, 100 μM, and 1 mM NaI.

**Monitoring insulin secretion**. Cell culture media were discarded just before collection for the measurement. Media (60 μL) were collected and incubated with refreshed media for 1 h at 37 °C, 5% $CO_2$. After 1-h incubation, an additional 60 μL media were collected. Collected media were placed on ice until ready to assay. For stimulation of pancreatic islets, 450 mg/dL glucose-containing media (instead of 100 mg/dL) were used. Insulin secretion was monitored by measuring concentration of insulin in the culture media using ELISA (Invitrogen; #KAQ1251) following a protocol provided by the company.

To monitor insulin secretion from pancreas-on-a-chip, we co-cultured PDECs in the top chamber and pancreatic islets in the bottom chamber. Base media for PDECs, advanced DMEM/F12, contains insulin, which can affect the concentration of insulin secreted by pancreatic islets in the bottom chamber. It was switched to DMEM (same as pancreatic islets media). Two chips were prepared, Chip A and Chip B, to employ agonist (10 μM FSK) or inhibitor (20 μM CFTR$_{inh-172}$) of CFTR channel on the PDECs. The chips were incubated at 37 °C for 1 h and 60 μL media were collected from the bottom chamber. FSK (Chip A) and CFTR$_{inh-172}$ (Chip B) were employed on the top chambers and the chips were incubated at 37 °C for 1 h. Sixty microliters of media was collected from the pancreatic islets on the bottom chamber. For Chip A, the media in the bottom chamber were switched to high-glucose-containing media (450 mg/dL) and the chip was incubated at 37 °C for 1 h. For Chip B, a combination of FSK and CFTR$_{inh-172}$ were added to the top chamber and the chip was incubated at 37 °C for 1 h. The chip was incubated with high-glucose-containing media at 37 °C for 1 h. Sixty microliters of media were collected from the pancreatic islets in the bottom chamber.

**Statistical analysis**. Data were derived from at least three independent replicates. The level of marginal significance, $p$-value, was calculated using two-tailed Student's $t$ test for pairwise comparison and one-way analysis of variance with Bonferroni adjustment for multiple variations. A $p$ value <0.05 was considered significant.

**Reporting summary**. Further information on research design is available in the Nature Research Reporting Summary linked to this article.

## Data availability

All experimental data relevant to graphs and images within the article and supplementary information are available from corresponding authors upon request.

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

## Acknowledgements

We thank Dr. Gail Pyne-Geithman for editing the manuscript. This research was supported by the National Institutes of Health (DK080834, DK093045, P30 DK117467), and CF Foundation grants to K.S.M. (MUN18F0) and A.P.N. (NAREN14XX0).

## Author contributions

K.S.M. performed all experiments and was assisted by K.A. (imaging, RNA-sequencing, and experimental design), Y.H. (experimental design),), F.Y. (RNA-sequencing), Y.R. (imaging), and B.N.A. (culturing pancreatic islets in vitro). K.S.M. fabricated the microfluidic device assisted by S.Y., J.D.N., and M.A.-E.-H., and J.J.P. performed the surgery, TPIAT. K.S.M., K.A., Y.H., M.A.-E.-H., J.D.N. J.J.P, and A.P.N wrote and edited the manuscript. A.P.N., J.D.N., and J.J.P. designed the project.

## Additional information

**Competing interests:** The authors declare no competing interests.

