## [Peer Review File · Nature Communications]

Reviewers' comments:

Reviewer #1 (Remarks to the Author):

The manuscript "Patient-derived pancreas-on-a-chip to model cystic fibrosis-related diabetes" submitted by Mun et al. describes the fabrication of pancreatic ductal organoids and a pancreas co-culture microfluidic device for the analysis of CFTR functionality and its relationship to insulin secretion. In their manuscript they demonstrated successful fabrication of pancreatic ductal organoids. Utilizing their co-culture microfluidic device they demonstrated that there could be a significant relationship between CFTR functionality in the epithelial cells and insulin control in the islet cells of the pancreas. Still, a majority of the paper is focused on the derivation of the cells and characterization of the pancreatic ductal organoids. There is a lack of experimentation to show the benefits of the microfluidic model and the CF-related diabetes that could be modeled. The manuscript would benefit from further examination of the interactions between CF epithelial cells and insulin regulation in context of understanding the mechanisms of CF-related diabetes. Further demonstration of the benefits of their microfluidic model and its capabilities would be useful as well as demonstration of the advantages of their approach for cell isolation methodology and formation of pancreatic ductal organoids. The authors show good insight into future experimentation that the design could be utilized for, but the current manuscript lacks impact and novelty.

Major comments:

1. The authors' hypothesis of the close proximity between islets of Langerhans and pancreatic ductal epithelial cells as the cause of CF Diabetes is not supported by any pathophysiological mechanism in the literature.
2. The methodology to form "organoids" from pancreatic ductal epithelial cells seems overly technical, involving multiple steps of dissection, manual passaging, and spheroid formation in Matrigel, followed by monolayer culture, and spheroid formation again. The authors do not show any benefit to this organoid formation strategy over standard cell isolation and culture methods. Could the cells be immediately utilized to form a monolayer? There is no evidence provided that these organoids show any functional improvement over standard primary cultures of cells. Is the only thing making these cultures organoids their 3D "spheroid" formation? If so what is the benefit of this?
3. The authors used tissue leftover from Islets isolation pancreata with background of acute pancreatitis; it is reported the presence of Islets and pancreatic ductal cells in the leftover tissue subsequently used by the authors to develop and manufacture the organoid to mimic CF-related diabetes. Overall the authors used cells with a background of acute pancreatitis (non CF), leftover pancreata used for the islet isolation underwent then subsequent processing in order to extrapolate a mixture of endo- and exocrine tissue used for the organoid formation highlighting a rather confused and inconsistent isolation protocol with an end product not mimicking a pancreas suffering from CF. Additionally, the experiment only inhibited CFTR channel function instead of utilizing a mutated CFTR model. The impact of these studies would be improved if CF pancreatic cells were obtained or if the CFTR gene was mutated and was phenotypically rescued by previously established drugs. Only one experiment was utilized to examine this relationship and the manuscript's impact with regards to this statement would be increased if further testing was completed that supported this conclusion.
 - a. The experiment to examine the relationship between CFTR function in PDECs and islet insulin functionality would benefit from a control experimentation to determine if there is any effect of the CFTR inhibitor directly on the islet cells' ability to produce insulin.
 - b. Further metabolic experimentation such as quantification of the intermediates before the production of the insulin could aid in the statement
4. The microfluidic model is quite simplistic, and there is only a small amount of data in the paper characterizing it and providing experimental data. There is no data showing the benefit of using microfluidics. It seems that the co-culture of epithelial cells and islets is the novel approach of the manuscript, however the advantage of this approach is not demonstrated well. For example, there is only one experiment showing the relationship between CFTR functionality in the epithelial cells

to insulin production in islet cells. The one experiment run in the manuscript could most likely be replicated in a simple co-culture or transwell culture model by isolating the two cell types in the upper and lower chambers. In the discussion of the manuscript, the significance of the two vacuum chambers in the chip is expressed, but there is no experimentation to support that its addition provides unique significance. It is possible that it could play a role in studying pancreatic pressure for pancreatitis, but this is not supported in any of the results.

Minor comments:

1. Was there a methodology to confirm complete monolayers of the cells had formed for the experimentation such as transepithelial electrical resistance (TEER) or was it just visual examination?
2. ROCK inhibitor was utilized to aid in cellular recovery and proliferation in the methods, but does this affect the functionality of cell types that are being examined? Is there previous research support to show little effect or can further characterization experiments be completed? A brief literature review as shown that a different ROCK inhibitor utilized on beta-like cells expressed increased glucose-stimulated insulin secretion and affected the growth of glucose-responding cells.

Reviewer #2 (Remarks to the Author):

This manuscript describes an elegant chip model that could be useful to understand disease mechanisms. In the model they induce a change in the regulatory conductance channels that they relate to cystic fibrosis. An ultimate proof of the model would be to obtain patient material from cystic fibrosis patients that preserves the phenotypes in the model.

Reviewer #3 (Remarks to the Author):

The manuscript written by Mun et al. describes a pancreas-on-a-chip platform, which has a great potential to understand the interplay between human primary pancreatic islets and pancreatic duct epithelial cells for investigations on cystic fibrosis-related diabetes. The first 3 figures of the manuscript were devoted to the characterization of isolated PDECs and islets, which has been well investigated in the literature. The authors could move most of the data from these figures to the supplementary data. The very novel findings of authors were provided in Figs. 4 and 5, which are mostly on the pancreas-on-chip model, and further experiments should be investigated on the chip to further analyze the interplay between endocrine and exocrine groups, such as the long term presence of CFTR inhibition on islet functionality (glucose responsiveness to insulin) and viability of beta cells. Treated islets should be retrieved from the chip and further analyzed (protein and gene expression profiles, insulin granules in the cells, etc) and compared to the islets that are not treated. I find the paper very interesting and innovative, but not complete and needs further findings and major revision to be considered for publication.

- "have been used a scaffold for cell..." This statement is not correct. Scaffold and PDMS device are different.

- "...with narrowing diameters (Supplemental Figure 3)". I couldn't see the branches with narrowing diameters. There is a single change with some zigzags. It is not clear why the authors did not make it straight.

- The function of the vacuum chambers is not clear. What is role of mechanical pressure to the cell culture chamber?

- Is it not clear if the statistical analysis approach is appropriate. ANOVA should be followed by a post hoc test.

- Scale bars in Fig. 1 h-j are not clear. The terminology on Figs. 1h-j is awkward, i.e., "isolation organoids," "re-forming organoids". Growing similar organoids using iPS derived lung epithelial cells in Matrigel turned into organoids with lumen inside. Perhaps the authors should back this up with published work as this behavior was observed previously in epithelial cells.

- Fig. 5 a does not match with the H&E in Fig. 5c. Epithelial lumen and islet architecture does not resemble to the one shown in H&E image. Epithelial marker expression is not strong though.

- Paragraph beginning with "Pancreatic ductal..." needs a ref.

- X-axis on Fig. 4d was not provided, should be Days?
- If pancreatic islets of 21 patients were isolated, why did the authors use only 3 samples for the analysis on the chip? This is the most important part of the manuscript with a huge deviation from patient to patient, as shown in Fig. 5e. Islets from health donors should be used as control in islet characterization to make sure the ones from pancreatitis are not impaired.
- Immunostaining in Fig. 3d is not convincing. Insulin is not uniformly distributed and the image of the islet was partially provided. Some cells were stained to insulin and glucagon together. N=3 is low due to the fact that there might be significant deviation from patient to patient.
- As there is a 10-micron filter between the islets and monolayer, the cross talk between both layers depends on the diffusion of molecules such as hormones and proteins to the other site. How do the islets sit on the chip? At the very bottom? What are their proximity the epithelial barrier? In the native pancreas, epithelial tube directly locates next to the islets but in the chip there is a separator. No discussion was provided on this.
- The authors also need to confirm the epithelial barrier function and structure on the chip, such as TEER and immunostaining to tight junctions, as some were done in Fig. 2, previously.
- The conclusion from Chip A in Fig. 5E is not clear. High glucose without FSK should be treated as well to describe the role of activation of CFTR channel.
- In general, the results and discussion done on Fig. 5E, particularly Chip B, is poorly described. This is the most important part of the manuscript and should be explained and discussed properly.

We thank the reviewers for their constructive comments and we have performed all the experiments and believe that the manuscript in its current form is greatly enhanced for publication in Nature Communications.

Reviewers' comments:

Reviewer #1 (Remarks to the Author):

The manuscript "Patient-derived pancreas-on-a-chip to model cystic fibrosis-related diabetes" submitted by Mun et al. describes the fabrication of pancreatic ductal organoids and a pancreas co-culture microfluidic device for the analysis of CFTR functionality and its relationship to insulin secretion. In their manuscript they demonstrated successful fabrication of pancreatic ductal organoids. Utilizing their co-culture microfluidic device they demonstrated that there could be a significant relationship between CFTR functionality in the epithelial cells and insulin control in the islet cells of the pancreas. Still, a majority of the paper is focused on the derivation of the cells and characterization of the pancreatic ductal organoids. There is a lack of experimentation to show the benefits of the microfluidic model and the CF-related diabetes that could be modeled. The manuscript would benefit from further examination of the interactions between CF epithelial cells and insulin regulation in context of understanding the mechanisms of CF –related diabetes. Further demonstration of the benefits of their microfluidic model and its capabilities would be useful as well as demonstration of the advantages of their approach for cell isolation methodology and formation of pancreatic ductal organoids. The authors show good insight into future experimentation that the design could be utilized for, but the current manuscript lacks impact and novelty.

Major comments:

1. The authors' hypothesis of the close proximity between islets of Langerhans and pancreatic ductal epithelial cells as the cause of CF Diabetes is not supported by any pathophysiological mechanism in the literature.

- Response: Cystic Fibrosis-Related Diabetes (CFRD) is a common lethal complication observed in CF-patients. The risk of diabetes in CF patient increases over time. CFRD is present in about 2% of children, 19% of adolescents and 40~50% of adults. Some literature has argued that the reason of CFRD being observed over time is due to lost mass of the islet Langerhans from pancreatic fibrosis^{1, 2}. As argued by Efrat and Russ in their review³, that ductal cells are in close proximity to islets (See Figure A below; from their review), similarly we observed that the islet cells are in close contact with the basolateral membrane of ductal epithelial cells as shown in Figure 5A and 5C (of the manuscript). Additionally we observed CFTR is only expressed in the pancreatic ductal epithelial cells, not in islet cells⁴. Our data, Figure 5E, shows that defective CFTR function leads to reduced insulin secretion in islet cells. In our humble opinion, these observations demonstrate a direct correlation between cystic fibrosis and onset of diabetes.

Figure A: Pancreatic ductal epithelial cells are located in close proximity to islet cells

2. The methodology to form “organoids” from pancreatic ductal epithelial cells seems overly technical, involving multiple steps of dissection, manual passaging, and spheroid formation in Matrigel, followed by monolayer culture, and spheroid formation again. The authors do not show any benefit to this organoid formation strategy over standard cell isolation and culture methods. Could the cells be immediately utilized to form a monolayer? There is no evidence provided that these organoids show any functional improvement over standard primary cultures of cells. Is the only thing making these cultures organoids their 3D “spheroid” formation? If so what is the benefit of this?

- Response: Primary cells have contamination issue with other cell types and it has been shown that the ductal cells can get contaminated easily by other cell types including acinar, fibroblasts and endocrine cells⁵. In order to avoid the contamination, during isolation of pancreatic ductal epithelial cells (PDECs) from pancreatic remnant cell pellet followed by Total Pancreatectomy with Islet Autotransplantation (TPIAT), are dissected carefully to avoid any other tissue/cell contamination. If isolated cells are cultured directly in a cell culture dish, PDECs will be contaminated with other cell types especially fibroblasts. One of the benefits of organoid culture is that we can manually pick pancreatic ductal organoids and culture pure PDECs. Because the pancreatic ductal organoids enlarge over time such a sphere is easy to isolate and avoid contaminating cells further. Also our RNAseq data shows that the preparation is quite pure (Fig 2i).

For freezing, we cultured PDECs on plastic dishes, and then transferred them directly to cryopreservation vial. We observed that survival during thawing cells was reduced dramatically, if the cells were frozen directly from the organoids; therefore, we freeze them from cultured dishes. Revival of PDECs from frozen vial can be cultured on a cell culture dish directly as monolayer, however PDECs tend to differentiate and stop proliferating over a period of time in the plate. Therefore, we re-formed organoids in Matrigel and then transferred them to filters or Chip to differentiate. These organoids maintain stem like feature and the data is highly reproducible including CFTR function (e.g., using chamber or Iodide efflux studies). Both organoids obtained from the fresh tissue, the monolayer of PDECs, and frozen PDECs showed similar basal secretion in the luminal area (please see Fig 1h, 1i, and 1j in the manuscript).

3. The authors used tissue leftover from Islets isolation pancreata with background of acute pancreatitis; it is reported the presence of Islets and pancreatic ductal cells in the leftover tissue

subsequently used by the authors to develop and manufacture the organoid to mimic CF-related diabetes. Overall the authors used cells with a background of acute pancreatitis (non CF), leftover pancreata used for the islet isolation underwent then subsequent processing in order to extrapolate a mixture of endo- and exocrine tissue used for the organoid formation highlighting a rather confused and inconsistent isolation protocol with an end product not mimicking a pancreas suffering from CF. Additionally, the experiment only inhibited CFTR channel function instead of utilizing a mutated CFTR model. The impact of these studies would be improved if CF pancreatic cells were obtained or if the CFTR gene was mutated and was phenotypically rescued by previously established drugs. Only one experiment was utilized to examine this relationship and the manuscript's impact with regards to this statement would be increased if further testing was completed that supported this conclusion.

a. The experiment to examine the relationship between CFTR function in PDECs and islet insulin functionality would benefit from a control experimentation to determine if there is any effect of the CFTR inhibitor directly on the islet cells' ability to produce insulin.

- Response: We thank the reviewer for the suggestion and we have performed these studies. We cultured islet cells only in the bottom chamber and applied the CFTR inhibitor, CFTR_{inh-172}, in the top chamber. We measured insulin concentration from the bottom chamber as shown in the Figure B. We did not observe any effect of the CFTR inhibitor by itself on insulin secretion.

Figure B: Effect of CFTR inhibitor to insulin secretion

b. Further metabolic experimentation such as quantification of the intermediates before the production of the insulin could aid in the statement

- Response: Very recently (1/3/2019) we obtained pancreatic remnant cell pellet from pancreatitis patient who was also diagnosed with CF and underwent TPIAT. The patient has dF508 (allele 1) and R117H (allele 2). The patient was also heterozygote for SPINK1 mutation. This patient was diagnosed to have mild CF and has some CFTR function as demonstrated by the mild phenotype (i.e., BMI: 19.84, sweat chloride: 51mmol/L, FEV1 predicted: 114% and is not diabetic). Our data also demonstrates that the ducts epithelial organoids show partial CFTR

function (see Fig C). The normal range of sweat chloride is less than 29 mmol/L and the value considered as CF is over 60 mmol/L (40-60 is considered partial function mutations).

We isolated pancreatic ductal organoids and examined CFTR function using fluid secretion assay. It was compared with the data obtained from 21 pancreatitis patients in the Figure 3a in the original manuscript. The organoids isolated from CF patient responded to the forskolin (FSK), which is an activator for cAMP-mediated CFTR channel opening. This patient's CFTR function (basal secretion) was 20% lower than non-CF pancreatitis patients as shown in the Figure C. The fluid secretion response increased to 70% upon stimulation by FSK (near normal range; although slightly lower than non-CF ducts, see Fig C).

We isolated islet cells from the patient and examined endocrine function using ELISA. We switched culture media from low glucose-containing media (100 mg/dL) to high glucose-containing media (450 mg/dL) as shown in the Figure D. The islet cells were stimulated and secreted more insulin upon incubation with high concentration glucose. The islet cells obtained from the pancreatitis/CF patient (mutants: delta F508, R117H, and SPINK1) showed efficient endocrine function.

We co-cultured pancreatic ductal epithelial cells and islet cells in the same chip isolated from the pancreatitis/CF patient and measured insulin concentration from the bottom chamber during inhibition of CFTR channel function in pancreatic ductal epithelial cells as the same condition in the manuscript. We normalized the absolute values of the secreted insulin into folds change as shown in the Figure E. We observed similar trend of insulin secretion during treatment with FSK and/or CFTR_{inh-172} from both patients, pancreatitis/Non-CF patient (54% decreased) and pancreatitis/CF patient (46% decreased) as shown in the Figure F.

We have examined metabolites (Metabolon Inc- performed these studies and we are still in the process of repeating these experiments) from secretions from apical and basolateral membrane of PDECs obtained pancreatitis/non-CF patient as shown in the Figure G below. These results are preliminary and would like the reviewer to know that we are being extremely cautious with this analysis. PDECs were cultured on a trans-well membrane and media from apical side and basolateral side was collected in the beginning and after incubation for 24 hours. We observed several metabolites were secreted significantly in 24 hours. Three of them (more than 15 folds) were closely related to insulin secretion. We are yet to confirm these results, but we will investigate the effect of these metabolites to regulate insulin secretion in islet cells; however we think it is beyond the scope of this manuscript.

Figure C: A fluid secretion assay to monitor CFTR function (*p-values*: ** < 0.005, **** < 1.0 x 10⁻⁵)

Figure D: Endocrine function in Islet cells of pancreatitis/CF patient (*p-value* < 0.02; n=3)

Figure E: Comparison insulin secretion between Pancreatitis/Non-CF and Pancreatitis/CF patient

Figure F: Comparison CFTR inhibitable insulin secretion

Figure G: Analysis of metabolites secreted from PDECs

4. The microfluidic model is quite simplistic, and there is only a small amount of data in the paper characterizing it and providing experimental data. There is no data showing the benefit of using microfluidics. It seems that the co-culture of epithelial cells and islets is the novel approach of the manuscript, however the advantage of this approach is not demonstrated well. For example, there is only one experiment showing the relationship between CFTR functionality in the epithelial cells to insulin production in islet cells. The one experiment run in the manuscript could most likely be replicated in a simple co-culture or transwell culture model by isolating the two cell types in the upper and lower chambers. In the discussion of the manuscript, the significance of the two vacuum chambers in the chip is expressed, but there is no experimentation to support that its addition provides unique significance. It is possible that it could play a role in studying pancreatic pressure for pancreatitis, but this is not supported in any

of the results.

- Response: The pancreas-on-a-chip, double-channel chip, was innovated from a simplistic single-channel chip (microfluidic device) to generate functional pancreas *in vitro* that mimics human/*in vivo* physiology. This device, i.e., pancreas-on-a-chip is complex structure with two cell culture chambers which are separated by a thin layer of porous membrane, where two different cell types are lined as an opposite site. It was designed to be able to mimic not only pancreatic structure, but also those functions with highly sensitive manner. The volume of single chamber is 3.76 mm^3 meaning that we need only $3.76 \text{ }\mu\text{L}$ to fill in the chamber. The pancreas-on-a-chip was fabricated using polydimethylsiloxane (PDMS), which is transparent and flexible. It allowed us to build the vacuum chamber to apply mechanical pressure to mimic pancreatic pressure, e.g., Pancreatic pressure has been reported significantly higher in CP than in controls (29.9 ± 3.1 vs 7.2 ± 1.1 mmHg, $p < 0.001$)⁶ and an *in vivo* model of acute pancreatitis was developed by increasing intrapancreatic duct pressure (from 7~11 to 25~33 mmHg)⁷. The pancreas-on-a-chip is easily portable and we are able to set up perfusion system with controlling flow rate in the cell culture chamber. It allows us to study drug delivery and drug discovery in a quick response on a microscopy in a patient specific manner. It is our expectation that this device (pancreas-on-a-chip) will help to develop personalized medicine to treat CFRD (and pancreatitis).

It is to be noted that our focus in this manuscript is to investigate the functional correlation between PDECs and islet cells causing CFRD. But we believe this *in vitro* model system might inspire other researchers to study other diseases and physiological aspect that affects the endocrine and exocrine of the pancreas.

Minor comments:

1. Was there a methodology to confirm complete monolayers of the cells had formed for the experimentation such as transepithelial electrical resistance (TEER) or was it just visual examination?

- Response: Our apologies, we did not fully explain this aspect of the study. We have performed TEER on filters and also on pancreas-on-a-chip (data shown below. See Fig. N). In addition, H&E staining and immunostaining of tight junction (ZO-1) was used to confirm monolayer of the cells (see Fig H and O). Additionally, we measured the transepithelial electrical resistance (TEER) of polarized PDECs on a trans-well membrane using epithelial volt-ohm meter. When the TEER was over $1000 \text{ }\Omega/\text{cm}^2$, the filter was mounted in an Ussing chamber to monitor CFTR function.

Figure H: Monolayer of PDECs were verified using H&E stain and Immunofluorescent microscopy (ZO-1; tight junction)

2. ROCK inhibitor was utilized to aid in cellular recovery and proliferation in the methods, but does this affect the functionality of cell types that are being examined? Is there previous research support to show little effect or can further characterization experiments be completed? A brief literature review as shown that a different ROCK inhibitor utilized on beta-like cells expressed increased glucose-stimulated insulin secretion and affected the growth of glucose-responding cells.

- Response: To address this concern, we applied ROCK inhibitor, Y-27632 (10 μ M), to the RPMI 1640 media on the first day only of the isolation. Islet cells were incubated with low glucose-containing media without ROCK inhibitor the next day and the media was changed every other day. In order to monitor endocrine function using the pancreas-on-a-chip after co-culturing of pancreatic ductal epithelial cells and islet cells, we cultured islet cells on a plate for at least one week and transferred them on the bottom chamber of the chip. The ROCK inhibitor did not have any effect on insulin secretion. We repeatedly observed efficient endocrine function of the islet cells isolated from the pancreatic remnant cell pellet. We examined the affect of Y-27632 as acute (Figure I), but we did not observe any significant change in insulin secretion in the presence of ROCK inhibitor.

Figure I: Effect of ROCK inhibitor, Y27632, to insulin secretion

Reviewer #2 (Remarks to the Author):

The manuscript is an important contribution to body or human on-a-chip model and how these can contribute to understand disease mechanisms.

1. the conclusion of the paper is that the patient derived pancreatic ductal cells and islet cells from the same patient is functional and it is possible to study cross-talk between the cell types. How well are the cells in the chip representing the phenotype of the patient? Will cells from a Cystic fibrosis patient represent a disease state in the chip?

- Response: Yes, we can study the cross talk between PDECs and islet cells using pancreas-on-a-chip in vitro model system as described in the comment 3b (Reviewer #1) and comment 14 (Reviewer #3). We isolated cells from pancreatitis patients, who had normal sweat chloride values (see Table below; Patient 2 & 4-7), which is an index of normal CFTR function and is consistent with data from cultured PDECs. From these ductal organoids we observed normal CFTR mediated fluid secretion *in vitro* as shown in Figure C. We obtained islet cells from

pancreatitis patient who was also diagnosed with Cystic Fibrosis (non-diabetic). We observed normal endocrine function *in vitro* (single channel chip: Fig 4-manuscript; double-channel chip: Fig B, 24-well plate: Fig D). These observations suggest that the cells appear to maintain *in vitro* the phenotype observed in the patients.

2. The study describes an elegant way to isolate and pancreatic ductal epithelial cells. The ductal cells are cultured, cryopreserved before they were applied in the chip. Related to question 1: How well is the phenotype of the ductal cells preserved? It would be important to provide phenotypic information, such as proteomic profile to understand whether the cells represent the subject it is originating from.

- Response: Based on the RNA sequencing in pancreatic ductal organoids cultured in Matrigel, we observed pure pancreatic ductal epithelial cells population without contamination from blood cells, acinar cells, or islet cells (see Fig 2I). We also verified using immunostaining that the pancreatic ductal organoids and monolayer of pancreatic ductal epithelial cells in culture express canonical epithelial cell specific markers (E-cadherin, ZO-1, ENaC, KRT 19, and CFTR) (see Fig 2 and Fig O).

3. There is little information on the human subjects. A table describing the subjects in supplementary information would be useful.

- Response: We thank the reviewer for the suggestion. We have added the patient information in Table 1 (supplementary material).

Table 1. TPIAT patient summary

Patients	Mutations		Gender	Age (years)	Sweat	FEV1 (%)	BMI
	Allele 1	Allele 2					
Patient 1	SPINK1 (pN34S)	None	F	14	ND	ND	34
Patient 2	CFTR (Δ 508 Het)	None	M	8	2	ND	17
Patient 3	None	None	M	13	ND	ND	23
Patient 4	PRSS1 (R122H)	None	F	9	16	ND	18
Patient 5	CFTR (1454G>C Het)	None	F	4	21	ND	15
Patient 6	CFTR (R170H)	SPINK1 (pN34S)	M	18	19	ND	21
Patient 7	None	None	F	13	46	108	25
Patient 8	CPA1	None	F	13	ND	ND	28
Patient 9	CFTR (Δ 508), SPINK1	CFTR (R117H)	F	15	51	114	20

4. Figure 5 describes a key experiment. On page 10, line 217 it is described that FSK increase insulin secretion by 16.9 %. But this is not statistical significant? So could the author describe this as a change?

- Response: Thank you for the comment, we have modified this statement in the main manuscript. Although in the presence of FSK, there was a small trend of increased insulin secretion from the islets, it was not statistically significant (see Fig E). Importantly, we observed that FSK by itself did not have any effect on insulin secretion.

5. Also in Fig 5 the effect of CFTR inhibitors will decrease the insulin concentration in the medium. The authors describe this as an effect on the ductal cells. There should be a proper control to study the effect on islets only by the CFTR inhibitor.

- Response: We performed new experiments in islets cells that were cultured by itself without any PDECs to monitor the effect of CFTR inhibitor on insulin secretion from islet cells independent of PDECs. We observed that the CFTR inhibitor did not have any effect on insulin secretion from islet cells only culture (Fig B).

6. On page 5, line 147 it is described that basal secretion of the ductal organoids was 60% before treatment and increased to 75% upon incubation with FSK. The sentence is unclear what the percentage number are referring to (what is 100%?)

- Response: The fluid secretion was calculated by the ratio of luminal volume to entire volume (please see equation below);

$$\text{Fluid secretion (\%)} = (\text{Luminal volume of the organoid} / \text{Total volume of the organoid}) * 100$$

In theory, a 100% secretion would indicate that the organoid lumen overlapped with the total organoid volume. In the fluid secretion assay in this study, fluid secretion will always be < 100% using the above described method of measurement.

Reviewer #3 (Remarks to the Author):

The manuscript written by Mun et al. describes a pancreas-on-a-chip platform, which has a great potential to understand the interplay between human primary pancreatic islets and pancreatic duct epithelial cells for investigations on cystic fibrosis-related diabetes. The first 3 figures of the manuscript were devoted to the characterization of isolated PDECs and islets, which has been well investigated in the literature. The authors could move most of the data from these figures to the supplementary data. The very novel findings of authors were provided in Figs. 4 and 5, which are mostly on the pancreas-on-chip model, and further experiments should be investigated on the chip to further analyze the interplay between endocrine and exocrine groups, such as the long term presence of CFTR inhibition on islet functionality (glucose responsiveness to insulin) and viability of beta cells. Treated islets should be retrieved from the chip and further analyzed (protein and gene expression profiles, insulin granules in the cells, etc) and compared to the islets that are not treated. I find the paper very interesting and innovative, but not complete and needs further findings and major revision to be considered for publication.

1. "have been used a scaffold for cell..." This statement is not correct. Scaffold and PDMS device are different.

- Response: Our apologies we have removed that statement. The new statement reads "In recent decades, microfluidics have been used as an *in vitro* model system for cell culture".

2. "...with narrowing diameters (Supplemental Figure 3)". I couldn't see the branches with narrowing diameters. There is a single change with some zigzags. It is not clear why the authors did not make it straight.

Figure J: Structure of cell culture chamber of microfluidic device

- Response: We have generated both patterns: straight channel and angled (10°) to mimic branches. Based on the ductal structure (Fig. 1e) we thought the branches with narrowing diameter would be more appropriate. The main cell culture channels (purple; solid lines) are branches as shown in the figure J above. Extended channels (red; dot lines) were cut and cell culture channels were connected from wide (1 mm) to narrow (0.656 mm) with reducing 10% diameter at each angle to form a single channel. It is to mimic single pancreatic duct. The pattern needs to be modified more precisely in terms of the length, diameter, and angle of human pancreatic duct in nature. However, this branched pattern with narrowing diameters might provide proof of concept that different location in pancreatic duct has different features such as different flow rate and pH level⁸.

3. The function of the vacuum chambers is not clear. What is role of mechanical pressure to the cell culture chamber?

- Response: The pancreas-on-a-chip was fabricated using polydimethylsiloxane (PDMS), where the material is flexible and gas permeable. These features are unique to mimic pathological pancreatic function *in vitro* such as pancreatic pressure. Pancreatic pressure has been reported significantly higher in CP than in controls (29.9 ± 3.1 vs 7.2 ± 1.1 mmHg, $p < 0.001$)⁶. Acute pancreatitis *in vivo* model has been successfully induced by increasing intrapancreatic duct pressure (from 7~11 to 25~33 mmHg)⁷.

4. It is not clear if the statistical analysis approach is appropriate. ANOVA should be followed by a post hoc test.

- Statistical analysis was performed using two-tailed Student's t-test for pairwise comparison and one-way ANOVA with Bonferroni adjustment for the p-value.

5. Scale bars in Fig. 1 h-j are not clear. The terminology on Figs. 1h-j is awkward, i.e., "isolation organoids," "re-forming organoids". Growing similar organoids using iPS derived lung epithelial cells in Matrigel turned into organoids with lumen inside. Perhaps the authors should back this up with published work as this behavior was observed previously in epithelial cells.

- Response: Our apologies, we have revised Figure 1 and is shown in the Figure K below. We observed patient-derived pancreatic ductal organoid formed duct-like structure and PDECs migrating out from the organoid to form a monolayer. Based on this observation, we breakdown the Matrigel and transferred manually picked-pancreatic ductal organoids to a new plate before forming the duct-like structure. This process reduced formation time from the organoids to monolayer. iPS derived lung epithelial cells cannot make duct-like structure.

Figure K: Revised Figure 1 in the manuscript

6. Fig. 5 a does not match with the H&E in Fig. 5c. Epithelial lumen and islet architecture does not resemble to the one shown in H&E image. Epithelial marker expression is not strong though.

- Response: Immunofluorescence image (Fig. 5a) was the same region of the H&E image (Fig. 5c) obtained from serial sectioned slide. The Figure 5a was apart approximately 30 ~ 50 μm from the Fig. 5c (we received 15 slides of the same position: 5 μm /slide). The green circle (CFTR; Fig 5a) indicates cross sectioned luminal area of the pancreatic ducts.

7. Paragraph beginning with “Pancreatic ductal...” needs a ref.

- Response: Thank you , we have included the references ‘Pancreatic ductal epithelial cells (PDECs) are reported to have the highest expression of CFTR in the body’^{9, 10, 11}.

8. X-axis on Fig. 4d was not provided, should be Days?

- Response: Our apologies for an oversight; X-axis indicates Time (min) on Fig 4d.

9. If pancreatic islets of 21 patients were isolated, why did the authors use only 3 samples for the analysis on the chip? **This is the most important part of the manuscript with a huge deviation from patient to patient, as shown in Fig. 5e. Islets from health donors should be used as control in islet characterization to make sure the ones from pancreatitis are not impaired.**

- Response: It took us a long time to master the techniques of consistent isolation, chip fabrication and importantly generate fully functional pancreas-on-a-chip. Using the first 11 patients, we reproducibly isolated pancreatic ductal epithelial cells from the pancreatic remnant cell pellet and characterized using RNA sequence and immunofluorescent microscope with epithelial cell markers. Additionally, we verified the function of pancreatic ductal epithelial cells by monitoring CFTR activity using fluid secretion and short-circuit current assay. From the 12th patient onwards, we isolated islet cells from the pancreatic remnant cell pellet of the same patient followed by TPIAT. We examined the islet cells using immunofluorescence staining of insulin and glucagon and monitored endocrine function *in vitro*. Before we co-cultured PDECs and islet cells in the double-channel chip, we verified cell specific functions including CFTR function of PDECs and endocrine function of islet cells with a small number of cells using single-channel chip.

We compared endocrine function of islet cells from non-pancreatic (and non-CF) disease patient and pancreatitis patient as shown in the Figure L. In both islet cell type, there was a significant increase in insulin secretion upon exposure to high glucose-containing media. It tells us that endocrine function of islet cells from pancreatitis patient is robust and not impaired (n=3). More recently we have generated chips from four additional patients.

Figure L: Compared insulin secretion in islet cells from normal patient and pancreatitis patient

10. Immunostaining in Fig. 3d is not convincing. Insulin is not uniformly distributed and the image of the islet was partially provided. Some cells were stained to insulin and glucagon together. N=3 is low due to the fact that there might be significant deviation from patient to patient.

- Response: Thank you, we have replaced the original figure the following figure M. Islet cells were verified using dithizone staining during isolation from the pancreatic remnant cell pellet. Islet cells were additionally characterized using insulin and glucagon specific immunostaining, following isolation.

Figure M: Immunofluorescent microscopy of insulin and glucagon

11. As there is a 10-micron filter between the islets and monolayer, the cross talk between both layers depends on the diffusion of molecules such as hormones and proteins to the other site. How do the islets sit on the chip? At the very bottom? What are their proximity the epithelial barrier? In the native pancreas, epithelial tube directly locates next to the islets but in the chip there is a separator. No discussion was provided on this.

- Response: Islet cells sit on the porous membrane. For co-culture of PDECs and islet cells, PDECs were seeded in the top chamber and cultured until the PDECs attached on the porous membrane (this process typically takes a few hours to 24 hours). Next, islet cells were seeded in the bottom chamber, the chip was, then, flipped over for culturing islet cells on the membrane directly (this process takes typically 6 hours). The gap between PDECs and islet cells is 10 μm corresponding to the thickness of the porous membrane. In the native pancreas, pancreatic ductal epithelial cells are covered by connective tissue and collagen (Fig. 5c, in the manuscript) and the distance can vary (i.e., between duct and islets). The distance range between PDECs and islet cells are a few microns to about 10 microns.

12. The authors also need to confirm the epithelial barrier function and structure on the chip, such as TEER and immunostaining to tight junctions, as some were done in Fig. 2, previously.

- Response: We thank the reviewer for this suggestion and we have performed these experiments as shown below. PDECs were cultured on the top chamber of the pancreas-on-a-chip and measured transepithelial electrical resistance (TEER) using epithelial volt-ohm meter at day 8 after seeding the cells as shown in the Figure N. The chopstick electrodes were connected to Ag/AgCl wires (0.5 mm diameter) and inserted into the tubing to measure the resistance between top and bottom chambers. We observed 1042 Ω/cm^2 of TEER value from the PDECs in the pancreas-on-a-chip. It was very close to the value (1055 Ω/cm^2) obtained from trans-well membrane to monitor CFTR function using Ussing chamber. The polarized monolayer of PDECs in the top chamber of the pancreas-on-a-chip were examined by immunofluorescence staining with DAPI (nucleus) and ZO-1 (tight junction) as shown in the Figure O.

Figure N: Transepithelial electrical resistance (TEER) from the pancreas-on-a-chip

Figure O: Z-stack of immunofluorescent microscope, ZO-1 and DAPI

13. The conclusion from Chip A in Fig. 5E is not clear. High glucose without FSK should be treated as well to describe the role of activation of CFTR channel.

- Response: The occurrence of diabetes increases over time (i.e., as they get older) in CF patient, however the mechanism is still unclear. One underlying possibility of developing CFRD is pancreatic fibrosis which destroys insulin-producing islet cells resulting in inefficient endocrine function^{1,2}. Our approach is based on the data demonstrating that we do not observe CFTR in islets, and we find that some portion of PDECs are in close proximity to islet cells which is the basolateral side of PDECs (almost in contact in some cases; see Fig 5a and 5c in the manuscript). We hypothesized that cell-cell functional interaction between the two cell types may play a role in CFRD and indeed that was observed.

Endocrine function of islet cells is directly affected by glucose stimulation. We observed that insulin secretion from the islet cells could be dramatically stimulated upon exposure to high concentration glucose. Activation or inhibition of CFTR function did not stimulate insulin secretion from islet cells not co-incubated with PDECs (Figure B).

14. In general, the results and discussion done on Fig. 5E, particularly Chip B, is poorly described. This is the most important part of the manuscript and should be explained and discussed properly.

- Response: As per the reviewer suggestions, we have rewritten this part for clarity. From the chip B, we observed insulin secretion was significantly decreased by inhibition of CFTR channel in the PDECs obtained from pancreatitis patient. More recently (1/3/2019) we obtained PDECs and islet cells from pancreatitis/CF patient (deltaF508/R117H) who had partial CFTR function (see Fig C) but was not diabetic. We isolated PDECs and islet cells from the patient and co-cultured both cells in the pancreas-on-a-chip to monitor insulin secretion as before (see Fig E). We observed similar trend that inhibition of CFTR function in PDECs led to a dramatically decrease in the insulin secretion of islet cells from both patients, pancreatitis/Non-CF patient and pancreatitis/CF patient.

Our data obtained from pancreas-on-a-chips suggests that CFTR plays a role in the regulation of insulin secretion from the islet cells and dysfunctional CFTR due to CF mutations may contribute to CFRD.

- References

1. Uc A, *et al.* Chronic Pancreatitis in the 21st Century - Research Challenges and Opportunities: Summary of a National Institute of Diabetes and Digestive and Kidney Diseases Workshop. *Pancreas* **45**, 1365-1375 (2016).
2. Wooldridge JL, Szczesniak RD, Fenchel MC, Elder DA. Insulin secretion abnormalities in exocrine pancreatic sufficient cystic fibrosis patients. *Journal of cystic fibrosis : official journal of the European Cystic Fibrosis Society* **14**, 792-797 (2015).
3. Efrat S, Russ HA. Making β cells from adult tissues. *Trends in Endocrinology & Metabolism* **23**, 278-285 (2012).
4. Strong TV, Boehm K, Collins FS. Localization of cystic fibrosis transmembrane conductance regulator mRNA in the human gastrointestinal tract by in situ hybridization. *The Journal of clinical investigation* **93**, 347-354 (1994).
5. Reichert M, Takano S, Heeg S, Bakir B, Botta GP, Rustgi AK. Isolation, culture and genetic manipulation of mouse pancreatic ductal cells. *Nature protocols* **8**, 1354-1365 (2013).
6. Manes G, Büchler M, Piemmico O, Sebastiano PD, Malfertheiner P. Is increased pancreatic pressure related to pain in chronic pancreatitis? *International Journal of Pancreatology* **15**, 113-117 (1994).
7. Romac JMJ, Shahid RA, Swain SM, Vigna SR, Liddle RA. Piezo1 is a mechanically activated ion channel and mediates pressure induced pancreatitis. *Nature communications* **9**, 1715 (2018).
8. Cafilisch CR, Solomon S, Galey WR. In situ micropuncture study of pancreatic duct pH. *American Journal of Physiology-Gastrointestinal and Liver Physiology* **238**, G263-G268 (1980).
9. Marino CR, Matovcik LM, Gorelick FS, Cohn JA. Localization of the cystic fibrosis transmembrane conductance regulator in pancreas. *The Journal of clinical investigation* **88**, 712-716 (1991).
10. Wilschanski M, Novak I. The Cystic Fibrosis of Exocrine Pancreas. *Csh Perspect Med* **3**, (2013).
11. Saint-Criq V, Gray MA. Role of CFTR in epithelial physiology. *Cell Mol Life Sci* **74**, 93-115 (2017).

Reviewers' comments:

Reviewer #1 (Remarks to the Author):

The authors have made significant efforts to address many of the concerns raised in the initial review. However there still remain critical flaws that have not been addressed, as well as several assumptions not currently supported by the evidence. The authors have several claims and assumptions that they have relied upon for their overall experimental design that not supported by any direct evidence, which include;

1) That close proximity between islets of Langerhans and pancreatic ductal epithelial cells as the cause of CF Diabetes

2) That the complex methodology of organoid fabrication has any benefit over standard cell isolation and culture methods, other than a way to obtain pure pancreatic ductal epithelial cell cultures.

3) That the two vacuum chamber chip has any functionality or relevance for the presented data. While interesting in potentially modeling pancreatic pressure, no evidence in its utility is presented.

Additionally, while the authors are on the right track by generating CF organoids, the use of a single heterozygote sample with partial CFTR function and a mild-normal phenotype (patient predicted FEV1 of 114%) does not provide any conclusive findings. The authors need to perform these types of experiments with organoids from multiple CF organoids, ideally with d508 homozygote samples.

Reviewer #2 (Remarks to the Author):

The authors have responded to the queries in an acceptable way and the manuscript is now acceptable for publication

Reviewer #3 (Remarks to the Author):

The authors responded to most of my comments; however, there are still a few concerns. The authors need to revise the manuscript one more time to address these issues.

- Scale bars in Fig. 1h from Day 0 to 5 should be removed. Lung epithelial organoids in Matrigel generate lumen but not the duct. It can be included in the discussion that similar lumen formation has been observed in other organoids but the authors also showed the duct formation in vitro, which is new.

- TEER values mentioned in the revised text do not match with the values in their response. 1055 vs 1200?

- ZO-1/DAPI staining on the epithelial barrier (supplementary Fig 3) is not clear, DAPI is not visible on XY plane, and ZO-1 is not uniform. This makes the barrier function questionable.

- H&E and immunofluorescence in fig. 5a-c is not convincing, despite the authors explained what they did. CFTR staining is not strong and does not show the epithelium clearly. In addition, the highlighted area in the dashed rectangle has a relatively straight lumen of the duct but there are a few green stained points and their relative positions are not straight. 30 microns should not make this difference.

2ND ROUND, Reviewers comments

We are thankful to the editor and reviewers for their interest in our manuscript and for their critical comments to achieve the completeness and quality of the manuscript necessary for publication in *Nature Communications*.

Reviewer #1 (Remarks to the Author):

The authors have made significant efforts to address many of the concerns raised in the initial review. However there still remain critical flaws that have not been addressed, as well as several assumptions not currently supported by the evidence. The authors have several claims and assumptions that they have relied upon for their overall experimental design that not supported by any direct evidence, which include;

1) That close proximity between islets of Langerhans and pancreatic ductal epithelial cells as the cause of CF Diabetes

- Response: Please allow us to clarify; we have based our claims in the manuscript on the experimental evidence as follows: (i). By immunostaining we observe that CFTR in the duct is in close physical proximity to islets (See Fig. 5a in the manuscript and also Fig. A below). (ii). CFTR_{inh-172} has a significant inhibition on insulin secretion (Pancreas-on-a-chip) when they are in close proximity to each other on a double channel chip (Fig. 5e). (iii). this insulin secretion inhibition (using CFTR_{inh-172}) is completely eliminated when the chip does not have ductal epithelial cells on the double channel chip (Supplemental Fig. 9). (iv). Also, CF-patients appear to show reduced insulin secretion (Supplemental Fig. 10C and 10D) although not significant. We have also performed a new set of experiments to address this very concern about the need of proximity between islets of Langerhans and pancreatic ductal epithelial cells as the cause of CFRD. We generated a PDMS membrane lacking pores (so that there is no contact between ducts and islets); under such condition, we observed that CFTR_{inh-172} does not have an effect directly on insulin secretion (see Fig. B and C).

Figure A: Immunofluorescent microscopy of insulin and CFTR (white arrows) from head of pancreas (a) and remnant cell pellet following islet isolation (b). Scale bar: 20 μ m.

Figure B: Monitoring insulin secretion from islet cells on double-channel chip, in which there are no pores on the membrane.

Figure C: Comparison of insulin secretion in islet cells from three different conditions that had no pancreatic ductal epithelial cells on the apical side (CASE I; from supplemental Fig. 9a), lacked pores on the membrane in the double-channel chip (CASE II; from Fig. B above and supplemental Fig. 9b), or co-cultured pancreatic ductal epithelial cells and islet cells on the pancreas-on-a-chip (CASE III; from Supplemental Fig. 10c). Islet cells were incubated with low glucose-contained media (100 mg/dL) for 60 min before collecting media to monitor insulin secretion.

2) That the complex methodology of organoid fabrication has any benefit over standard cell isolation and culture methods, other than a way to obtain pure pancreatic ductal epithelial cell cultures.

- Response: The method is rather simple, except that we introduce an additional step of generating organoids. A major issue in isolation of primary cells from human tissue is contamination by other cell types (this is a major reason we chose to use organoids). The pancreatic remnant cell pellet following islet isolation process contains fibroblasts, acinar cells, islet cells, and pancreatic ductal epithelial cells. Fibroblast contamination is most problematic in 2D culture. Upon isolation of pancreatic ductal epithelial cells cultured through normal cell culture protocol, cell sorting is required such as fluorescence activated cell sorting (FACS); we find that this leads to an excessive loss of cells during the process (patient samples are precious and we try to obtain as many as cells possible). The pancreatic ductal organoids grow dramatically into large spheres over time, thereby enriching the ductal epithelial cell population significantly. Remarkably, we have been able to obtain organoids as large as 4.2 mm in diameter, which is visible to the naked eye, at day 27 from the isolation as shown in Figure D. We believe the inclusion of organoid isolation described in this manuscript provides: (a). Isolation of very pure population of ductal epithelial cells, (b). Robust expansion and (c) importantly, a highly reproducible method to consistently generate pure populations of ductal epithelial cells. (d). as well as the opportunity for easy freezing and reviving. It is to be noted that we have distributed these organoids to other labs (e.g., University of Iowa-Aliya Uc Lab) and they have been able to generate monolayers and monitor CFTR-dependent function.

Figure D. Patient-derived pancreatic ductal organoid.

3) That the two vacuum chamber chip has any functionality or relevance for the presented data. While interesting in potentially modeling pancreatic pressure, no evidence in its utility is presented.

- Response: We will refrain from referencing the vacuum chamber in this manuscript as it is not part of this study. However, it was our intention to design this novel pancreas-on-a-chip *in vitro* model system to mimic functional and structural attributes of the pancreas *in vivo*. The vacuum chambers in the chip have a unique function to apply pancreatic pressure to the cell culture chambers. We hope this elegant microfluidic device will allow the study of pancreatic related diseases for patients in future studies.

4) Additionally, while the authors are on the right track by generating CF organoids, the use of a single heterozygote sample with partial CFTR function and a mild-normal phenotype (patient

predicted FEV1 of 114%) does not provide any conclusive findings. The authors need to perform these types of experiments with organoids from multiple CF organoids, ideally with $\Delta F508$ homozygote samples.

- Response: We have obtained pancreatic remnant cell pellets from 22 pancreatitis patients who underwent TPIAT. Most of the patients (21 patients, 95%) were pancreatitis/non-CF patient and the last patient and the only patient so far (22nd patient) was a pancreatitis/CF patient (allele 1- $\Delta F508$ and allele 2-R117H). Pancreatitis is an infrequent complication among patients with cystic fibrosis¹. In 10,071 patients with CF from 29 different countries, De Boeck, et al. estimated overall occurrence of pancreatitis among patients with CF of 1.24%. Given these findings, it is very unlikely we will have CF-Pancreas in the near future (e.g. a severe CF phenotype like $\Delta F508/\Delta F508$ form of CF). In fact, in $\Delta F508/\Delta F508$ or other forms of severe CF, the patients are typically pancreatic insufficient as early as the first 1-2 years of life, and not at risk for pancreatitis. It is only in the milder types of CF when patients are pancreatic sufficient that there is risk for pancreatitis².

Reviewer #2 (Remarks to the Author):

The authors have responded to the queries in an acceptable way and the manuscript is now **acceptable for publication**

- Response: Thank you.

Reviewer #3 (Remarks to the Author):

The authors responded to most of my comments; however, there are still a few concerns. The authors need to revise the manuscript one more time to address these issues.

- Scale bars in Fig. 1h from Day 0 to 5 should be removed.

- Response: Thank you, we have removed scale bars as suggested (Figure 1h).

- Lung epithelial organoids in Matrigel generate lumen but not the duct. It can be included in the discussion that similar lumen formation has been observed in other organoids but the authors also showed the duct formation in vitro, which is new.

- Response: Thank you, we revised the discussion section and have included ductal formation from the pancreatic ductal organoids.

- TEER values mentioned in the revised text do not match with the values in their response. 1055 vs 1200?

- Response: Our apologies, please allow us to clarify. In the manuscript, we explained, "Trans-epithelial electrical resistance was measured using epithelial volt-ohm meter (World Precision Instruments, #EVOM and #STX2) and the trans-well membrane was mounted in an Ussing chamber when the resistance was over $1000 \Omega/\text{cm}^2$ ". The value of the resistance more precisely is $1055 \Omega/\text{cm}^2$ when normalized by area of the membrane. (350Ω divided by the area

of a trans-well membrane (0.33183 cm^2) is $1055 \Omega/\text{cm}^2$). We observed the resistance of ductal epithelial cells, $1200 \Omega/\text{cm}^2$, for the graph as shown in Figure E below (Figure 3b in manuscript).

Figure E. Monitoring CFTR function using short-circuit current assay.

- ZO-1/DAPI staining on the epithelial barrier (supplementary Fig 3) is not clear, DAPI is not visible on XY place, and ZO-1 is not uniform. This makes the barrier function questionable.

- Response: Thank you. We have replaced the supplementary Figure 3 with the following z-stack image (Fig. F).

Figure F: Z-stack of immunofluorescent microscopy of ZO-1 and DAPI in pancreatic ductal epithelial cells on a porous membrane in the pancreas-on-a-chip.

- H&E and immunofluorescence in fig. 5a-c is not convincing, despite the authors explained what they did. CFTR staining is not strong and does not show the epithelium clearly. In addition, the highlighted area in the dash rectangle has a relatively straight lumen of the duct but there are a few green stained points and their relative positions are not straight. 30 microns should not make this difference.

- Response: Please allow us to clarify; we examined paraffin sections of head of pancreas following H&E stain (a) and Masson's Trichrome stain (b) from the same position as shown in Figure G. In between these two conditions there were 15 additional cuts from the same paraffin block. The immunofluorescent microscopy of insulin and CFTR (c) is one of the sections. The pancreatic duct lined just below the islet cells as long straight lumen (a), however the duct was

apart from the islet cells and changed structure having small lumen after serial sections (b). The lumen area might be at a junction with another pancreatic duct. We also have other images and areas (please see Fig. A).

Figure G. Head of pancreas was examined using (a) H&E stain, (b) Masson's Trichrome stain, and (c) immunofluorescent microscopy of Insulin and CFTR obtained from the same position.

- References:

1. De Boeck K, Weren M, Proesmans M, Kerem E. Pancreatitis among patients with cystic fibrosis: Correlation with pancreatic status and genotype. *Pediatrics* **115**, (2005).
2. Ooi CY, Durie PR. Cystic fibrosis transmembrane conductance regulator (CFTR) gene mutations in pancreatitis. *J Cyst Fibros* **11**, 355-362 (2012).

REVIEWERS' COMMENTS:

Reviewer #1 (Remarks to the Author):

While the authors have made some efforts to address my concerns, the major flaws in the study are still not addressed. Based on the authors comments these are difficult to overcome, thus the manuscript remains critically flawed.

Specifically, the approach is overly complicated and does not appear to have any benefit over standard cell isolation and culture methods. The authors have agreed that their chip model has no benefit, so remains a 2D co-culture model at best. Finally, it is not clear what their model is actually achieving. If pancreatitis is such an infrequent complication among patients, then the need for a new complicated culture model has minor impact. It is still not demonstrated that they have achieved physiologically relevant modeling of CF pancreatitis or CFRD so there is no apparent application of their work. Based on these critical flaws this manuscript remains unsuitable for publication in this journal.

Reviewer #3 (Remarks to the Author):

The authors have responded to my comments.

3RD ROUND, Reviewers comments

Reviewer #1 (Remarks to the Author):

While the authors have made some efforts to address my concerns, the major flaws in the study are still not addressed. Based on the authors comments these are difficult to overcome, thus the manuscript remains critically flawed.

Specifically, the approach is overly complicated and does not appear to have any benefit over standard cell isolation and culture methods. The authors have agreed that their chip model has no benefit, so remains a 2D co-culture model at best. Finally, it is not clear what their model is actually achieving. If pancreatitis is such an infrequent complication among patients, then the need for a new complicated culture model has minor impact. It is still not demonstrated that they have achieved physiologically relevant modeling of CF pancreatitis or CFRD so there is no apparent application of their work. Based on these critical flaws this manuscript remains unsuitable for publication in this journal.

- Response: Please allow us to clarify. There are multiple reasons we developed a pancreas-on-a-chip, which has several advantages over cell culture models:

(i) Compared to conventional human cell culture model, the chip allows fluid flow (by setting perfusion system in a cell culture incubator or on a microscope), relevant mechanical cues in cellular signaling, allows tissue-tissue interface (i.e., duct-islet interface) and separate access to the two tissues types studied here (i.e., duct and islets). Controlling fluid flow will allow us in the future to study delivery of drugs from various surfaces, and to monitor the outcome in real time to perform various biochemical assays and potential fluorescent live imaging.¹

(ii) It allows us to work with small volume of media, as only 3.7 μ L is needed to fill in each of the chambers. The thickness and volume of the cell culture chambers are 140 μ m and 3.7 mm³. It also requires very small number of cells (typically 1/10th of what is needed for the smallest filter available) and we can use the precious human samples judiciously.

(iii) It allows us to monitor CFTR function in pancreatic ductal epithelial cells and endocrine function in islet cells at the same time (also once adapted, in real time).

(iv) low cost materials to fabricate the double channels chip (\$1.37 per chip) comparing to commercially available trans-well membrane (CAT# 3470 from corning; \$4.85/each). The chip was composed of two cell culture layers fabricated by PDMS (CAT# 4019862 from Ells Worth Adhesive; \$1.21/chip) and a thin layer of porous membrane (CAT# 9480 from Momentive; \$0.16/chip).

Moreover, a fabrication of the chip is highly reproducible. Once a pattern is created (i.e., mold) on a silicon wafer using photolithography technique, this mold can be used indefinitely.

We are working very closely with the Pancreas Care Center team at our institution and this collaboration made it evident to us that while pancreatitis is a relatively rare entity, when it occurs, it leads to extreme pain and debilitation in the growing child, and results in frequent hospitalizations and morbidity. It is very much needed to create systems in the lab that help us understand mechanisms leading to pancreas destruction and complications related to pancreatitis, such as diabetes through models like our model. This is one way we hope to test therapies for this disease in the future. Recent NIH workshops have demonstrated the need for

such models to help develop therapies or even to understand the mechanisms of pancreatogenic diabetes and pancreatitis.^{2, 3, 4}

Reviewer #3 (Remarks to the Author):

The authors have responded to my comments.

- Response: Thank you.

References:

1. Ingber DE. Developmentally inspired human 'organs on chips'. *Development* **145**, 1-4 (2018).
2. Abu-El-Haija M, *et al.* Accelerating the Drug Delivery Pipeline for Acute and Chronic Pancreatitis: Summary of the Working Group on Drug Development and Trials in Acute Pancreatitis at the National Institute of Diabetes and Digestive and Kidney Diseases Workshop. *Pancreas* **47**, 1185-1192 (2018).
3. Forsmark CE, *et al.* Accelerating the Drug Delivery Pipeline for Acute and Chronic Pancreatitis: Summary of the Working Group on Drug Development and Trials in Chronic Pancreatitis at the National Institute of Diabetes and Digestive and Kidney Diseases Workshop. *Pancreas* **47**, 1200-1207 (2018).
4. Uc A, *et al.* Chronic Pancreatitis in the 21st Century - Research Challenges and Opportunities: Summary of a National Institute of Diabetes and Digestive and Kidney Diseases Workshop. *Pancreas* **45**, 1365-1375 (2016).